# DsbA-L deficiency in T cells promotes diet-induced thermogenesis through suppressing IFN-γ production

Haiyan Zhou [1,4✉], Xinyi Peng[1,4], Jie Hu[1], Liwen Wang[1], Hairong Luo[1], Junyan Zhang[1], Yacheng Zhang[1], Guobao Li[1], Yujiao Ji[1], Jingjing Zhang [1], Juli Bai[2], Meilian Liu [3], Zhiguang Zhou[1] & Feng Liu [1,2✉]

Adipose tissue-resident T cells have been recognized as a critical regulator of thermogenesis and energy expenditure, yet the underlying mechanisms remain unclear. Here, we show that high-fat diet (HFD) feeding greatly suppresses the expression of disulfide-bond A oxidoreductase-like protein (DsbA-L), a mitochondria-localized chaperone protein, in adipose-resident T cells, which correlates with reduced T cell mitochondrial function. T cell-specific knockout of DsbA-L enhances diet-induced thermogenesis in brown adipose tissue (BAT) and protects mice from HFD-induced obesity, hepatosteatosis, and insulin resistance. Mechanistically, DsbA-L deficiency in T cells reduces IFN-γ production and activates protein kinase A by reducing phosphodiesterase-4D expression, leading to increased BAT thermogenesis. Taken together, our study uncovers a mechanism by which T cells communicate with brown adipocytes to regulate BAT thermogenesis and whole-body energy homeostasis. Our findings highlight a therapeutic potential of targeting T cells for the treatment of over nutrition-induced obesity and its associated metabolic diseases.

[1] National Clinical Research Center for Metabolic Diseases, Metabolic Syndrome Research Center, and Department of Metabolism and Endocrinology, The Second Xiangya Hospital of Central South University, 410011 Changsha, Hunan, China. [2] Department of Pharmacology, University of Texas Health Science Center at San Antonio, San Antonio, TX 78229, USA. [3] Department of Biochemistry and Molecular Biology, University of New Mexico Health Sciences Center, Albuquerque, NM 87131, USA. [4] These authors contributed equally: Haiyan Zhou, Xinyi Peng. ✉email: hyzhou02@csu.edu.cn; liuf@uthscsa.edu

Obesity is caused by increased energy intake and/or decreased energy expenditure, the latter is determined by basal metabolism, physical activity, and adaptive thermogenesis[1]. Adaptive thermogenesis is induced by calorie excess or cold exposure, referred to as diet-induced thermogenesis (DIT) and cold-induced thermogenesis (CIT), respectively. Both types of adaptive thermogenesis are sensed by the brain, and rely on the activation of β3 adrenergic receptor (β3AR) signaling and uncoupling protein 1 (UCP1) expression[2,3]. Whereas the mechanisms underlying CIT have been extensively studied, much less is known about the mechanism regulating DIT. Over the past decade, several immune cell types such as Group 2 innate lymphoid cells (ILC2s), eosinophils, alternatively activated macrophages (AAMacs), and invariant natural killer T (iNKT) cells have been found to regulate adaptive thermogenesis in brown adipose tissue (BAT) and beige fat[4–6], both of which are specialized in the dissipation of energy in the form of heat. However, the roles and the mechanisms of T cells in regulating adaptive thermogenesis remains unclear.

Recently, there is some evidence suggesting that T cells may also be involved in the regulation of adaptive thermogenesis[7]. Adipose-resident γδ T cells, which are stimulated by cold exposure, have been found to promote thermogenesis via secreting IL-17[8]. Another study has otherwise suggested that γδ T cells could promote sympathetic innervation by driving the expression of TGF-β1 in parenchymal cells via the IL-17 receptor C[9]. Besides, CD8+ T cell deficiency has been shown to promote beige fat development and energy expenditure[10]. On the other hand, activation of sympathetic tone by cold exposure, β3AR agonist CL316,243 treatment, or short-term high-fat diet (HFD) feeding all induce regulatory T (Treg) cells in BAT while the loss of Treg cells significantly suppresses BAT thermogenic capacity and lipolytic function[11]. However, the mechanisms by which adipose-resident T cells regulate energy homeostasis remain largely unknown.

Mitochondria are considered as the bioenergetic, biosynthetic, and signaling center that coordinates T cell fate decisions and regulates T cell function[12,13]. Mitochondrial homeostasis and function are regulated by many factors including nuclear and mitochondrial encoded genes, stress and hormone signaling pathways, transcription factors, and chaperone proteins[14,15]. Disulfide bond A oxidoreductase-like protein (DsbA-L) is a chaperone protein initially found in the matrix of rat liver mitochondria[16]. Liver-specific knockout of DsbA-L impairs hepatic mitochondrial functions and exacerbates HFD-induced hepatosteatosis[17]. Consistent with a protective role of DsbA-L in mitochondrial function, fat-specific knockout of DsbA-L promotes HFD-induced mitochondrial DNA (mtDNA) release into the cytosol, leading to the activation of the cGAS-STING pathway and inflammatory responses[18,19]. However, the role of DsbA-L in T cells remains in its infancy.

In the current study, we show that HFD feeding greatly suppresses DsbA-L expression in T cells, which correlates with reduced mitochondrial function in BAT-resident T cells. T cell-specific knockout of DsbA-L downregulates T cell mitochondrial function but protects mice from HFD-induced obesity and metabolic dysfunction. Mechanistically, DsbA-L deficiency in T cells decreases IFN-γ production, leading to enhanced protein kinase A (PKA) signaling and increased thermogenic gene expression, which contributes to increased BAT thermogenesis. Taken together, our results identify DsbA-L as a critical regulator of T cell mitochondrial function and cytokine production and reveal a potential mechanism underlying the crosstalk between BAT-resident T cells and adipocytes in the regulation of energy homeostasis.

## Results

**HFD feeding downregulates T cell mitochondrial function in BAT.** To explore the link between energy homeostasis and T cell function, we first examined the effect of over-nutrition on T cell mitochondrial mass, a biomarker for T cell mitochondrial function. HFD feeding significantly increased the mitochondrial mass of CD4+ and CD8+ T cells in mouse BAT but not inguinal white adipose tissue (iWAT) (Fig. 1a, b). To test whether the HFD-induced increase in mitochondrial mass is caused by the accumulation of impaired mitochondria, we used a combination of Mito-Tracker Green with Mito-Tracker Red (stain of mitochondrial membrane potential (MMP)) or mitoSOX to distinguish between respiring mitochondria and dysfunctional mitochondria[20]. By flow cytometry analyses, we quantified dysfunctional mitochondria (mitoSOX-positive cells or MMP-low cells) (Fig. 1c) and found that HFD feeding increased the mitochondrial defective MMP-low cells in CD4+ T cells and to a lesser extent in CD8+ T cells of BAT (Fig. 1d), but increased MMP of T cells in iWAT (Fig. 1e). These findings suggest that over-nutrition has a distinct effect on T cell mitochondrial function in mouse BAT and iWAT. Consistent with these findings, mitochondrial ROS levels, as determined by mitoSOX fluorescence, were increased in both CD4+ and CD8+ T cells in BAT but not in iWAT of HFD-fed mice compared to control mice (Fig. 1f, g), demonstrating that mitochondria in BAT-resident T cells are more sensitive to HFD feeding-induced stress compared to those in iWAT-resident T cells.

**DsbA-L is a critical regulator of T cell mitochondrial function.** Activation of the sympathetic nervous system (SNS) induces T cell reprogramming, which has been shown to contribute to HFD feeding-induced BAT thermogenesis[11]. However, how T cells play a role in this thermogenic process remains unclear. There is some evidence suggesting that SNS activation reduces T cell mitochondrial mass, MMP, and respiration function[21]. Interestingly, we found that the expression of DsbA-L, a mitochondrial-localized chaperone protein, was also significantly suppressed in T cells by β3AR agonist CL316,243 treatment (Fig. 2a), suggesting a potential role of DsbA-L in mitochondrial dysfunction induced by HFD feeding.

We recently found that HFD feeding downregulates the expression levels of DsbA-L in adipocytes and hepatocytes, leading to impaired mitochondrial function in these cells[17,18]. To determine whether downregulation of DsbA-L affects mitochondrial function in T cells, we generated T cell-specific DsbA-L knockout mice (DsbA-L^CD4-KO) by crossing DsbA-L floxed mice[17] with CD4-Cre mice. DsbA-L expression was specifically suppressed in CD3+, CD4+, and CD8+ T cells as demonstrated by both PCR and Western blot experiments (Supplementary Fig. 1a, b). DsbA-L ablation led to reduced oxygen consumption rates (OCR) in CD4+, CD8+, and CD3+ T cells (Fig. 2b and Supplementary Fig. 1c), without affecting the extracellular acidification rates (ECAR) (Supplementary Fig. 1d). DsbA-L deficiency in T cells also ablated ATP production and mitochondrial DNA content in both CD4+ and CD8+ T cells (Fig. 2c, d), suggesting an essential role for DsbA-L in T cell mitochondrial function. Furthermore, DsbA-L deficiency enhanced T cell mitochondrial fission while reduced mitochondrial calcium levels upon TCR stimulation (Fig. 2e–g). Given that fused mitochondrial is beneficial for oxidative phosphorylation (OXPHOS)[22] and that calcium influx is a critical regulator of mitochondrial function[23], these results further suggest that DsbA-L deficiency decreased T cell OXPHOS and mitochondrial function. DsbA-L deficiency in T cells had no significant effect on thymocyte numbers (Supplementary Fig. 1e), frequencies of CD4+ and CD8+ T cells in the spleen and thymus

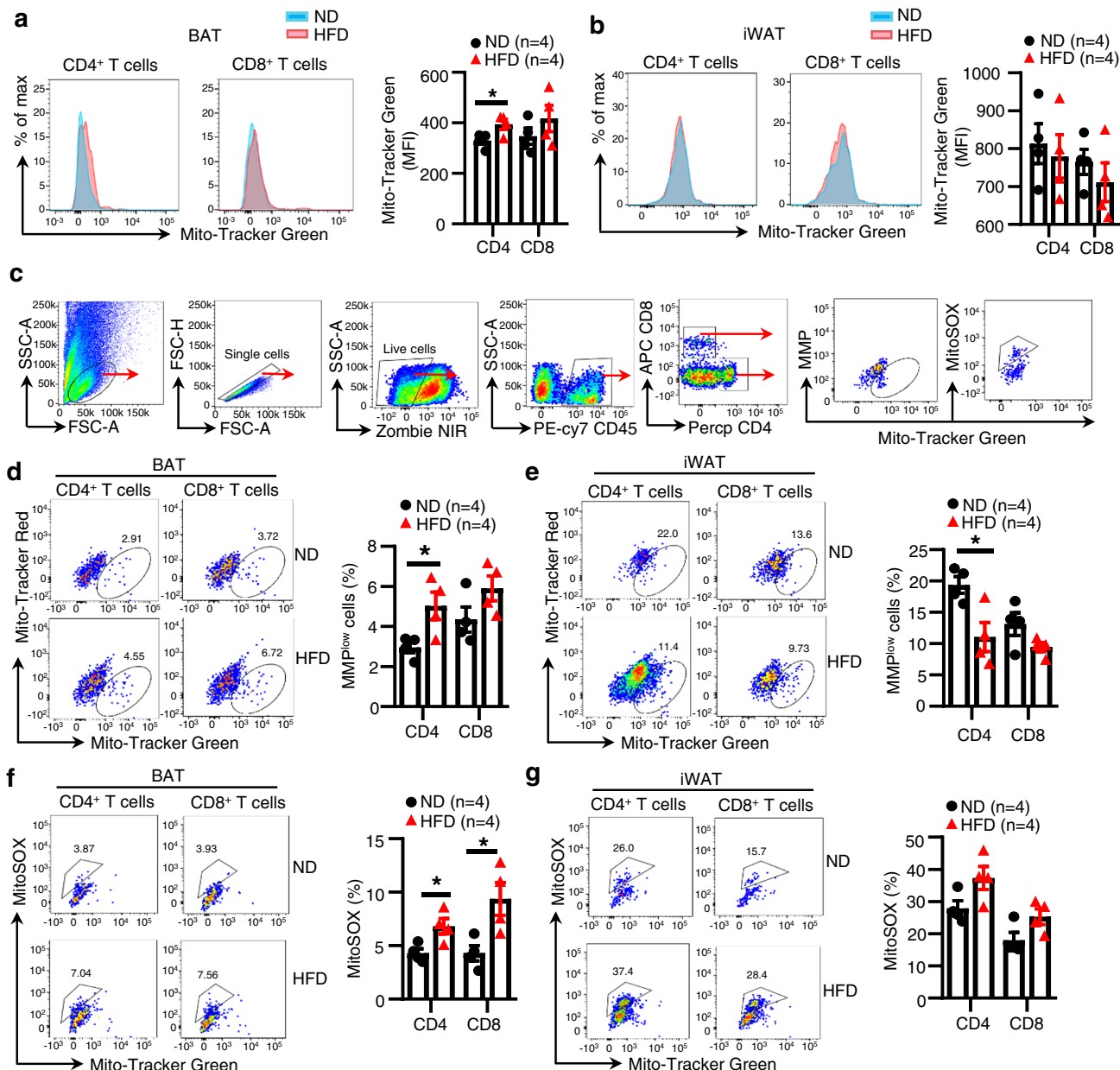

**Fig. 1 HFD feeding downregulates T cell mitochondrial function in BAT. a, b** Mitochondrial mass of T cells in BAT (**a**) and iWAT (**b**) from ND- and HFD-fed mice was analyzed by Mito-Tracker Green staining ($n = 4$/group). **c** Gating strategies of mitoSOX-positive and MMP$^{low}$ T cells in BAT and iWAT. **d, e** MMP of T cells in BAT (**d**) and iWAT (**e**) from ND- and HFD-fed mice was analyzed by Mito-Tracker Green and Mito-Tracker Red staining. ($n = 4$/group). **f, g** Mitochondrial ROS of T cells in BAT (**f**) and iWAT (**g**) from ND- and HFD-fed mice was analyzed by MitoSOX staining. ($n = 4$/group). Data shown are representative of three independent experiments. All data are presented as mean ± SEM. Statistical values $p < 0.05$ (*), $p < 0.01$ (**), $p < 0.001$ (***) are determined by two-tailed unpaired Student's $t$-test. Source data are provided as a Source Data File.

(Supplementary Fig. 1f), and thymic Treg development (Supplementary Fig. 1g). In addition, there was no significant difference in the percentages of naïve (CD62L+CD44$^{Low}$) versus memory (CD62L−CD44$^{High}$) T cell population in the spleen of both genotypes (Supplementary Fig. 1h). Taken together, these results demonstrate that DsbA-L is a key regulator of mitochondrial function in T cells, which are consistent with our previous findings that DsbA-L regulates mitochondrial function in adipocytes and hepatocytes[17,18].

**T cell-specific knockout of DsbA-L alleviates diet-induced obesity and insulin resistance**. To determine the potential role of

T cell-specific DsbA-L in adipose tissue function, DsbA-L$^{CD4-KO}$ mice and control littermates were fed a normal chow diet (ND) or an HFD for 12 weeks. Under ND feeding conditions, DsbA-L$^{CD4-KO}$ mice displayed no significant difference in body weight (Supplementary Fig. 2a), fat mass (Supplementary Fig. 2b), glucose tolerance (Supplementary Fig. 2c), and insulin sensitivity (Supplementary Fig. 2d) compared to their floxed control littermates. However, after challenging with an HFD for 12 weeks, DsbA-L$^{CD4-KO}$ mice displayed smaller body size (Fig. 3a), lower body weight (Fig. 3b), and reduced-fat mass (Fig. 3c) compared with control littermates. Consistent with these findings, the volumes (Fig. 3d) and relative tissue weights (Fig. 3e) of BAT, iWAT, perirenal fat

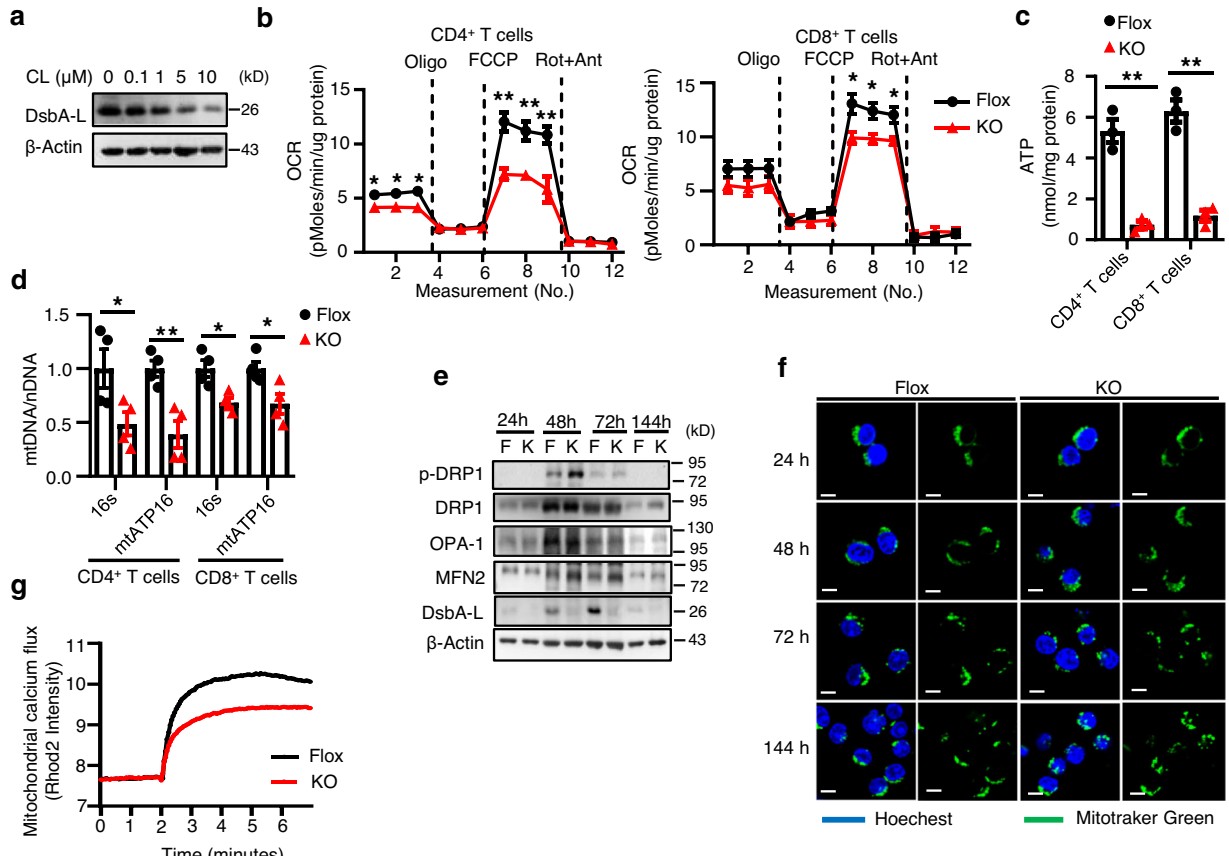

**Fig. 2 DsbA-L is a critical regulator of T cell mitochondrial function. a** DsbA-L expression in CD3+ T cells stimulated with different concentrations of CL316,243 for 24 h. **b** OCR of CD4+ ($n = 5$/group) and CD8+ ($n = 4$–5/group) T cells were measured under basal conditions and in response to indicated drugs. Oligo, oligomycin; Rot, rotenone; Ant, antimycin. **c** ATP production of activated CD4+ ($n = 3$/group) and CD8+ ($n = 3$/group) T cells isolated from the spleen of DsbA-L^CD4-KO mice and control littermates. **d** The relative mtDNA content in activated CD4+ ($n = 4$/group) and CD8+ ($n = 4$/group) T cells isolated from the spleen of DsbA-L^CD4-KO mice and control littermates. **e** Mitochondrial fission and fusion in activated CD4+ T cells isolated from the spleen of DsbA-L^CD4-KO mice and control littermates were measured by western blot analyses. **f** Mitochondrial morphology was evaluated at indicated time points using live-cell confocal microscopy after staining with 100 nM Mito-Tracker Green and 10 μg/ml Hoechst 33342. Scale bars: 5 μm. **g** Mitochondrial calcium flux was measured in activated CD4+ T cells isolated from the spleen of DsbA-L^CD4-KO mice and control littermates. Results are representative of three independent experiments. Data shown are representative of three independent experiments. All data are presented as mean ± SEM. Statistical values $p < 0.05$ (*), $p < 0.01$ (**), $p < 0.001$ (***) are determined by two-tailed unpaired Student's $t$-test. Source data are provided as a Source Data File.

(prWAT), and liver were significantly reduced in DsbA-L^CD4-KO mice compared with control littermates. In addition, the adipocyte size of iWAT was smaller in DsbA-L^CD4-KO mice compared with those in control littermates (Fig. 3f, g). Although T cell-specific knockout of DsbA-L had no significant effects on tissue volume and adipocyte size of epididymal white adipose tissue (eWAT) (Fig. 3d-g), the DsbA-L^CD4-KO mice showed reduced inflammatory gene expression in eWAT compared with control littermates (Supplementary Fig. 3a). Notably, T cell-specific knockout of DsbA-L markedly alleviated HFD-induced steatosis in the liver (Fig. 3d, f). DsbA-L^CD4-KO mice showed decreased fasting glucose levels and were more glucose tolerant compared with control littermates (Fig. 3h, i). While fasting insulin levels were not significantly different between DsbA-L^CD4-KO mice and control littermates, T cell DsbA-L deficiency significantly increased insulin sensitivity in mice (Fig. 3j, k). Consistent with these findings, insulin-stimulated AKT and GSK3β phosphorylation[24] was enhanced in BAT, iWAT, and eWAT of HFD-fed DsbA-L^CD4-KO mice compared with control littermates (Supplementary Fig. 3b–d). These results demonstrate that DsbA-L deficiency-induced T cell mitochondrial dysfunction has an important role in regulating whole-body energy homeostasis in mice.

**T cell-specific knockout of DsbA-L increases energy expenditure and BAT thermogenic function.** To elucidate the mechanism by which DsbA-L deficiency in T cells alleviates diet-induced obesity, we examined food intake, physical activity, intestinal absorption function, and energy expenditure between DsbA-L^CD4-KO mice and floxed control littermates fed an HFD for 12 weeks. T cell-specific knockout of DsbA-L had little effect on food intake (Supplementary Fig. 3e), physical activity (Supplementary Fig. 3f), or respiratory exchange ratio (RER) (Supplementary Fig. 3g) between the two genotypes. The HFD-fed DsbA-L^CD4-KO mice also showed comparable intestinal absorption function compared to control littermates as determined by feces weights and total fecal calorie (Supplementary Fig. 3h). Compared to control mice, the HFD-fed DsbA-L^CD4-KO mice showed a marked increase in UCP1 protein levels and thermogenic gene expressions in BAT (Fig. 4a–c) as well as a small increase in UCP1 protein and mRNA levels in iWAT (Fig. 4a, c). To determine whether the increased BAT thermogenesis contributes to the reduced obesity in DsbA-L^CD4-KO mice, we examined metabolic phenotypes in DsbA-L^CD4-KO mice and control littermates fed an HFD for just 5 weeks, a time point when DsbA-L^CD4-KO mice did not show any significant difference in body weight gain compared to control littermates (Fig. 3b). Indirect calorimetry

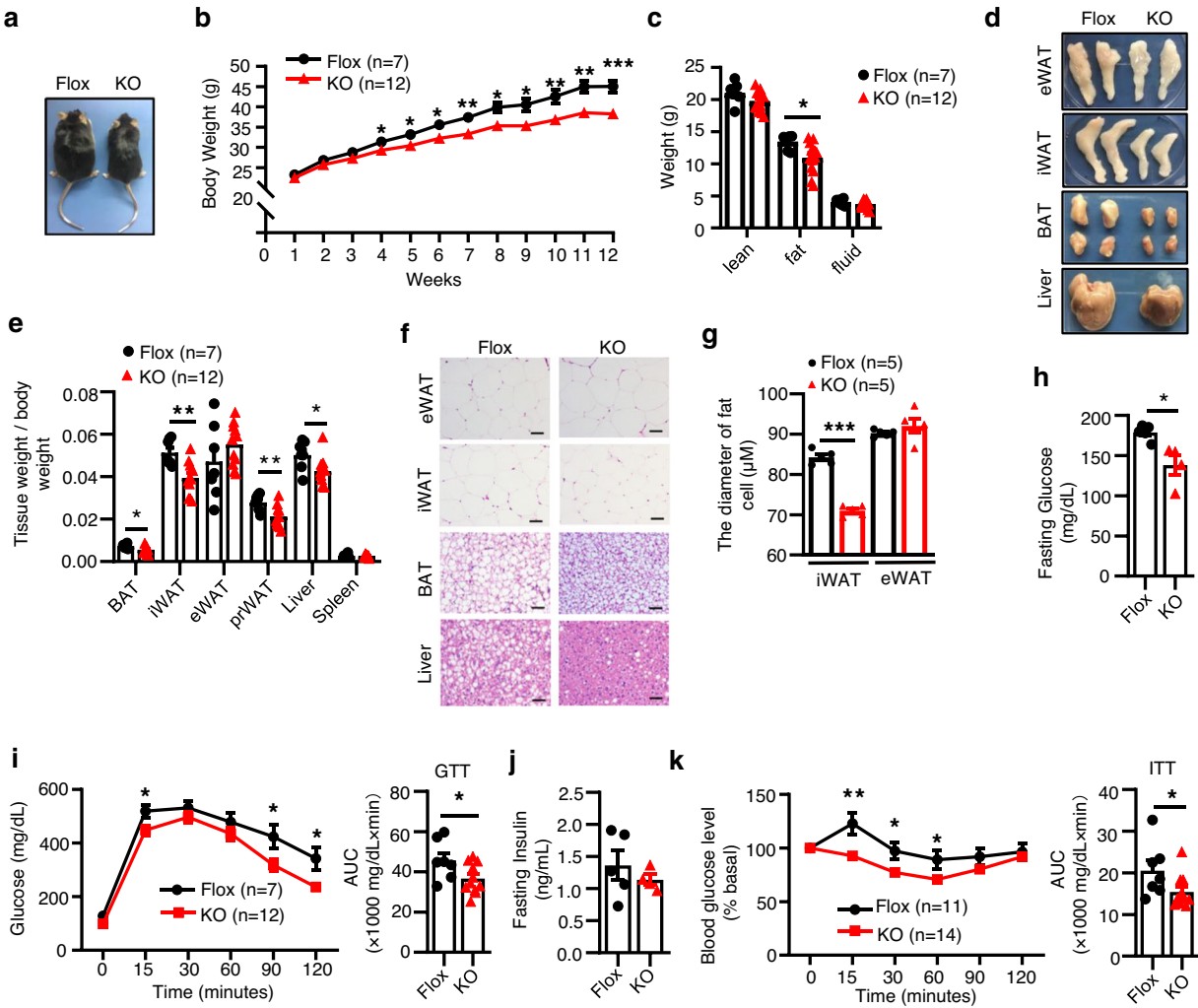

**Fig. 3 T cell-specific knockout of DsbA-L alleviates diet-induced obesity and insulin resistance. a** Representative images of mice after a 12-week HFD feeding regimen starting at 8 weeks of age. **b** Body weight gain of DsbA-L^CD4-KO mice ($n = 12$) and control littermates ($n = 7$) during HFD feeding. **c** Body composition of DsbA-L^CD4-KO mice ($n = 12$) and control littermates ($n = 7$) fed an HFD for 12 weeks. **d** Representative images of fat pads and liver collected from DsbA-L^CD4-KO mice and control littermates fed an HFD for 12 weeks. **e** Relative tissue weights of HFD-fed DsbA-L^CD4-KO mice ($n = 12$) and control littermates ($n = 7$) fed an HFD for 12 weeks. **f** Representative H&E staining of eWAT, iWAT, BAT, and liver of DsbA-L^CD4-KO mice and control littermates fed an HFD for 12 weeks. Scale bars: 50 μm. **g** The average diameters of fat cells in DsbA-L^CD4-KO mice and control littermates were analyzed and quantified by the Image J software. **h** Fasting glucose levels in DsbA-L^CD4-KO mice ($n = 4$) and control littermates ($n = 5$) fed an HFD for 12 weeks. **i** Glucose tolerance test was performed in DsbA-L^CD4-KO mice ($n = 12$) and control littermates ($n = 7$) fed an HFD for 12 weeks. **j** Fasting insulin levels in DsbA-L^CD4-KO mice ($n = 4$) and control littermates ($n = 5$) fed an HFD for 12 weeks. **k** Insulin tolerance test was performed in DsbA-L^CD4-KO mice ($n = 14$) and control littermates ($n = 11$) fed an HFD for 12 weeks. Data shown are representative of three independent experiments. All data are presented as mean ± SEM. Statistical values $p < 0.05$ (*), $p < 0.01$ (**), $p < 0.001$ (***) are determined by two-tailed unpaired Student's $t$-test. Source data are provided as a Source Data File.

experiments showed that HFD feeding increased DIT and the promoting effect was further enhanced in DsbA-L^CD4-KO mice, which displayed higher oxygen consumption and energy expenditure (Fig. 4d, e and Supplementary Fig. 3i). No significant differences in oxygen consumption and energy expenditure were observed between DsbA-L^CD4-KO mice and control littermates fed a ND (Fig. 4e). Importantly, BAT and iWAT collected from HFD-fed DsbA-L^CD4-KO mice also showed elevated basal OCR (Fig. 4f), suggesting that enhanced BAT and iWAT oxygen consumption may contribute to the increased whole-body energy expenditure in the DsbA-L^CD4-KO mice. Consistent with these findings, the expression of genes involved in mitochondrial biogenesis such as *Aco2*, *Atp5a1*, and *Sdhb* as well as fatty acid oxidation such as *Cpt1a*, *Cpt2*, *Mcad*, *Acox1*, and *Ppara*, were all increased in BAT, and to a less extent in

iWAT, of DsbA-L^CD4-KO mice compared to the control littermates (Fig. 4g). In addition, the protein levels of UCP1 and PGC1α were significantly increased in BAT of DsbA-L^CD4-KO mice compared to control littermates (Fig. 4h). Consistent with these findings, the mRNA levels of *Ucp1*, *Ppargc1a*, *Prdm16*, and *Cidea* were markedly upregulated in the BAT of DsbA-L^CD4-KO mice compared to the control littermates (Fig. 4i). Increased expression of a beige marker gene *Tbx1* was also observed in the iWAT of DsbA-L^CD4-KO mice compared to the control littermates (Fig. 4i). Interestingly, the HFD-fed DsbA-L^CD4-KO mice displayed enhanced oxygen consumption and energy expenditure under thermoneutral condition (30 °C) (Fig. 4j, k and Supplementary Fig. 3j), under which condition mice lack the thermal drive to activate brown or beige fat[3]. These results indicated that DsbA-L deficiency in T cells affects the intrinsic

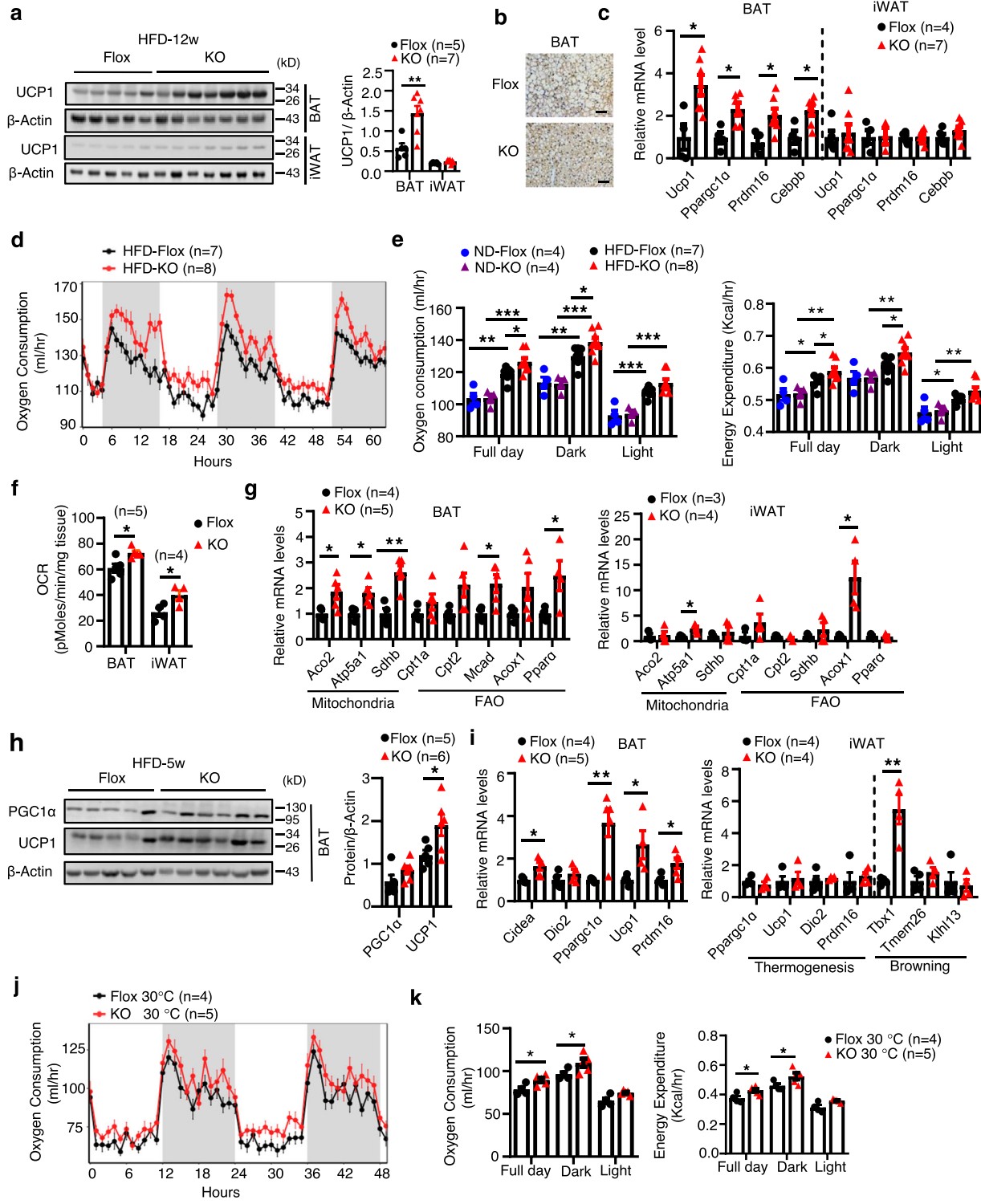

metabolic rate of the mice, which is consistent with the view that animals exhibited DIT even at thermoneutrality[3].

To determine whether T cell DsbA-L deficiency affects CIT in mice, singly housed DsbA-L[CD4-KO] mice and control littermates were kept at room temperature (25 °C) or cold temperature (6 °C) under a 12-h light/12-h dark cycle for 6 days. Under these conditions, comparable UCP1 protein expression levels were observed in BAT and iWAT between DsbA-L[CD4-KO] mice and control littermates (Supplementary Fig. 4a). Consistent with this finding, T cell-specific knockout of DsbA-L had no significant

effect on the mRNA levels of thermogenic genes such as *Ucp1*, *Ppargc1α*, and *Tbx1*, in BAT and iWAT of mice exposed to cold (Supplementary Fig. 4b, c). Taken together, these findings indicate that DsbA-L deficiency in T cells has a critical role in the diet- but not cold-induced BAT thermogenesis in mice.

## DsbA-L deficiency in T cells suppresses IFN-γ production while promoting Treg cell accumulation. To determine the mechanism by which adipose tissue-resident T cells regulate DIT,

**Fig. 4 T cell-specific knockout of DsbA-L increases energy expenditure and BAT thermogenic function. a** UCP1 expression in BAT and iWAT of DsbA-L[CD4-KO] mice ($n = 7$) and control littermates ($n = 5$) fed an HFD for 12 weeks. **b** Immunohistochemical staining of UCP1 in BAT of DsbA-L[CD4-KO] mice and control littermates fed an HFD for 12 weeks. Scale bars: 50 μm. **c** Thermogenic gene expression in BAT and iWAT of DsbA-L[CD4-KO] mice ($n = 7$) and control littermates ($n = 4$) fed an HFD for 12 weeks. **d** The oxygen consumption of DsbA-L[CD4-KO] mice ($n = 8$) and control littermates ($n = 7$) fed an HFD for 5 weeks was measured. **e** The average of oxygen consumption and energy expenditure between DsbA-L[CD4-KO] mice and control littermates fed an ND ($n = 4$/group) or HFD ($n = 7$–8/group) for 5 weeks. **f** The basal OCR of BAT ($n = 5$/group) and iWAT ($n = 4$/group) collected from DsbA-L[CD4-KO] mice and control littermates fed an HFD for 5 weeks. **g** mRNA levels of genes involved in mitochondrial and fatty acid metabolism in the BAT ($n = 4$–5/group) and iWAT ($n = 3$–4/group) of DsbA-L[CD4-KO] mice and control littermates fed an HFD for 5 weeks. **h** Western blot analyses of UCP1 and PGC1α in the BAT collected from DsbA-L[CD4-KO] mice ($n = 6$) and control littermates ($n = 5$) fed an HFD for 5 weeks. **i** mRNA levels of genes involved in thermogenesis in the BAT ($n = 4$–5/group) and iWAT ($n = 4$/group) of DsbA-L[CD4-KO] mice and control littermates fed an HFD for 5 weeks. **j** The oxygen consumption of DsbA-L[CD4-KO] mice ($n = 5$) and control littermates ($n = 4$), which were fed an HFD for 5 weeks and then tested under thermoneutral conditions (30 °C). **k** The average of oxygen consumption and energy expenditure between DsbA-L[CD4-KO] mice ($n = 5$) and control littermates ($n = 4$) which were fed an HFD for 5 weeks and then tested under thermoneutral conditions (30 °C). Data shown are representative of three independent experiments. All data are presented as mean ± SEM. Statistical values $p < 0.05$ (*), $p < 0.01$ (**), $p < 0.001$ (***) are determined by two-tailed unpaired Student's $t$-test (**a** (right panel), **c**, **f**, **g**, **h** (right panel), **i** or ANCOVA test (**e**, **k**). Source data are provided as a Source Data File.

we first examined the phenotypes of T cells in adipose tissues of DsbA-L[CD4-KO] mice and their control littermates fed an HFD for 12 weeks. Flow cytometry analyses of T cell subsets (Fig. 5a) revealed that DsbA-L deficiency in T cells had little effect on adipose-resident T cell populations, as demonstrated by similar frequencies and total numbers of CD4[+] and CD8[+] T cells between the DsbA-L[CD4-KO] mice and their control littermates (Supplementary Fig. 5a, b). However, the frequencies and total numbers of IFN-γ-producing Th1 cells and IFN-γ-producing CD8[+] T cells were significantly decreased in the BAT, and to a lesser extent in iWAT and eWAT, of DsbA-L[CD4-KO] mice compared to control mice (Supplementary Fig. 5c, d). On the other hand, the DsbA-L[CD4-KO] mice showed a moderate increase in the frequencies of Treg cells, but not Th2 cells, in the BAT when compared with control littermates (Supplementary Fig. 5e, f). These results suggest that DsbA-L deficiency induces reprogramming of T cells in adipose tissues after long-time overnutrition.

After HFD feeding for 5 weeks, the frequencies and total numbers of IFN-γ-producing Th1 cells and IFN-γ-producing CD8[+] T cells were all significantly decreased in the BAT of DsbA-L[CD4-KO] mice compared with control littermates (Fig. 5b, c). DsbA-L[CD4-KO] mice also displayed decreased frequencies of IFN-γ-producing Th1 cells in the iWAT, without affecting their total numbers (Fig. 5b). However, DsbA-L deficiency had no obvious effect on the frequencies and total numbers of Th2 cells in adipose tissues (Fig. 5d). Interestingly, we found that both the frequencies and total numbers of Treg cells, which are critical in promoting adipose tissue thermogenic function[11], were increased in the BAT of DsbA-L[CD4-KO] mice compared with control littermates (Fig. 5e). Nevertheless, there were no significant differences in the frequencies and total numbers of γδ T cells, eosinophils, and M2 macrophages, all of which have been reported to regulate BAT thermogenic function[5,8,11], between the two genotypes (Supplementary Fig. 5g–k). Consistent with the decrease of IFN-γ-producing T cells, a selective and significant decrease in IFN-γ mRNA expression was observed in the stromal vascular fractions (SVFs) of BAT from DsbA-L[CD4-KO] mice compared with their control littermates fed an HFD for 5 weeks (Fig. 5f). Decreased IFN-γ protein levels were also observed in BAT but not in iWAT of DsbA-L[CD4-KO] mice compared to control mice (Fig. 5g). No significant differences in circulating IFN-γ levels were observed between the two genotypes of the mice (Fig. 5h), suggesting a potential paracrine action of the IFN-γ produced by adipose-resident T cells.

Despite a modest decrease in the frequencies of IFN-γ[+]Th1 cells in the BAT of DsbA-L[CD4-KO] mice compared to control mice under cold stimulation conditions, (Supplementary Fig. 6a–g),

much lower frequencies of IFN-γ[+]Th1 cells were found in BAT of mice exposed to cold compared to mice fed an HFD (Fig. 5b, Supplementary Figs. 5c, 6a). The mRNA levels of IFN-γ were similar between the two genotypes in response to cold stimulation (Supplementary Fig. 6h). These findings provide an explanation as to why CIT was comparable between DsbA-L[CD4-KO] mice and control littermates. Taken together, these findings suggest DsbA-L deficiency affects the subsets of adipose-resident T cells differently, which decreases the abundance of IFN-γ-producing Th1 and CD8[+] T cells while increases the frequencies of Treg cells, especially in BAT. The DsbA-L deficiency-induced reprogramming of T cells may further contribute to altered metabolic phenotypes in mice after HFD feeding.

**IFN-γ inhibits thermogenic gene expression in brown adipocytes.** Increased Treg cells have been shown to promote thermogenic gene expression in BAT and iWAT[11]. However, the role of IFN-γ in the regulation of thermogenesis and energy expenditure remains elusive. Numerous studies have shown that IFN-γ expression in iWAT is quite low under lean conditions while it is greatly induced after HFD feeding[25]. However, whether HFD feeding promotes IFN-γ expression in BAT remains unclear. To address this question, we first examined UCP1 expression in mice fed an ND or HFD for 2, 4, and 12 weeks. Consistent with previous findings[26], HFD feeding increased the mRNA levels of UCP1 in BAT (Supplementary Fig. 7a) but decreased its expression in iWAT (Supplementary Fig. 7b). Interestingly, the mRNA levels of IFN-γ were greatly decreased in the BAT of HFD mice (Supplementary Fig. 7c). This negative association between UCP1 and IFN-γ expression in the BAT of HFD-fed mice suggests that IFN-γ may inhibit BAT thermogenesis.

To determine whether reduced IFN-γ levels contribute to increased thermogenic gene expression in BAT of the DsbA-L[CD4-KO] mice, we treated primary brown adipocytes with conditioned medium (CM) collected from explant cultured BAT of HFD-fed DsbA-L[CD4-KO] mice and control littermates. Primary brown adipocytes treated with CM of BAT explant from HFD-fed DsbA-L[CD4-KO] mice, which showed a significant decrease in IFN-γ levels compared to that from HFD-fed control mice (Fig. 6a), led to a noticeable higher UCP1 expression in cells compared to those treated with CM from HFD-fed control littermates (Fig. 6b, c). Notably, the increased thermogenic gene expression can be reversed by the addition of IFN-γ to the CM (Fig. 6b, c), suggesting an inhibitory role of IFN-γ in thermogenic gene expression in BAT.

To further dissect the mechanism by which IFN-γ suppresses thermogenesis, we examined the activation of the β3AR signaling pathway, which is known to be responsible for both CIT and

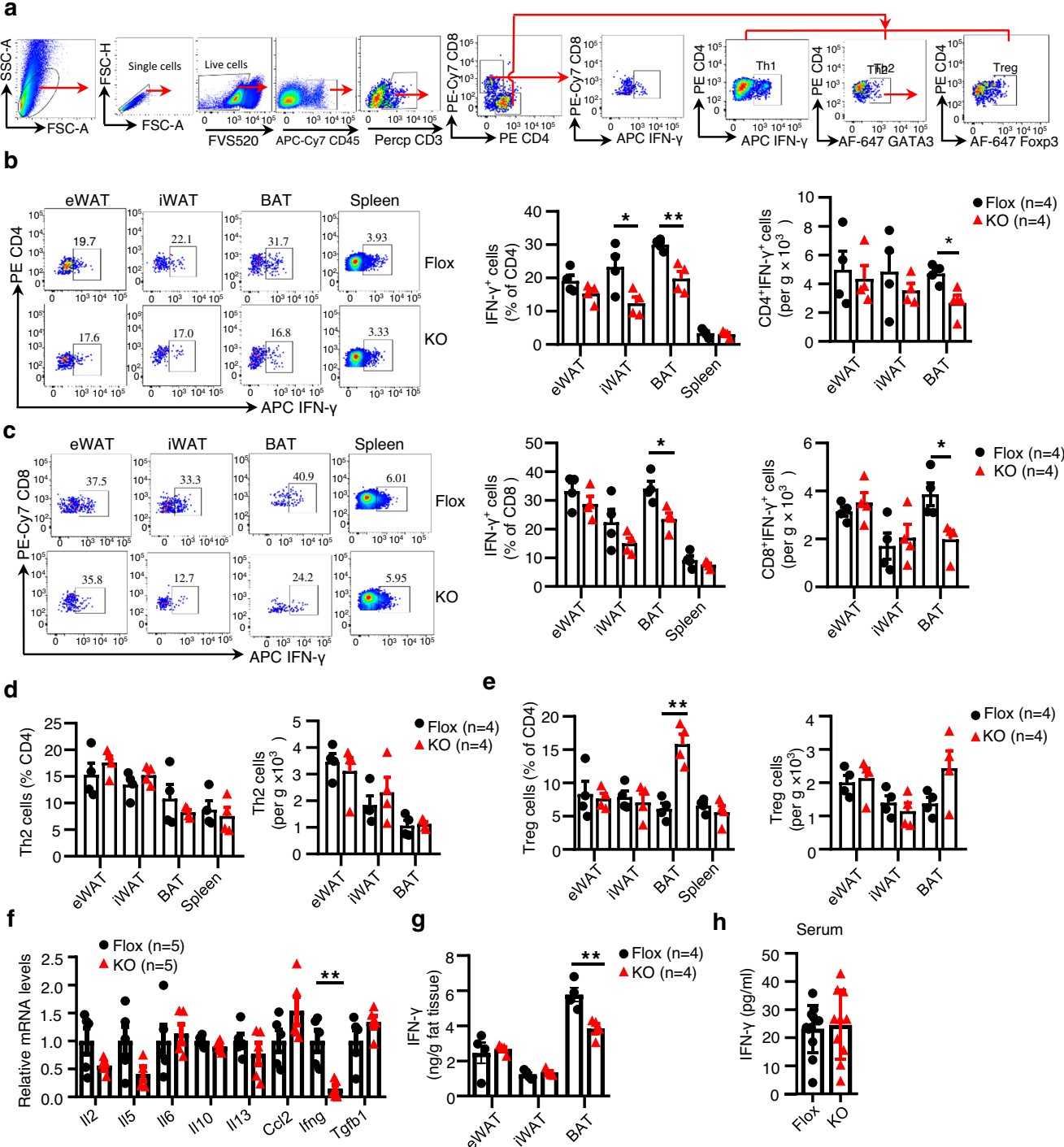

**Fig. 5 DsbA-L deficiency in T cells suppresses IFN-γ production while promoting Treg cell accumulation.** DsbA-L$^{CD4-KO}$ mice and control littermates were fed an HFD for 5 weeks. **a** Gating strategies of T cell subsets in adipose tissues. **b** Representative FACS plots of CD4$^+$IFN-γ$^+$ Th1 cells (left panel) and quantification of the frequencies and total numbers of CD4$^+$IFN-γ$^+$ Th1 cells (right panel). ($n = 4$/group). **c** Representative FACS plots of CD8$^+$IFN-γ$^+$ T cells (left panel) and quantification of the frequencies and total numbers of CD8$^+$IFN-γ$^+$ T cells (right panel). ($n = 4$/group). **d**, **e** Quantification of the frequencies and total numbers of Th2 cells (**d**) and Treg cells (**e**). ($n = 4$/group). **f** mRNA levels of T cell-associated cytokines in the SVF of BAT. ($n = 5$/group). **g** Protein levels of IFN-γ in the BAT and iWAT. ($n = 4$/group). **h** Protein levels of IFN-γ in the serum of DsbA-L$^{CD4-KO}$ mice ($n = 10$) and control littermates ($n = 12$). Data shown are representative of three independent experiments. All data are presented as mean ± SEM. Statistical values $p < 0.05$ (*), $p < 0.01$ (**), $p < 0.001$ (***) are determined by two-tailed unpaired Student's $t$-test. Source data are provided as a Source Data File.

DIT[3]. The expression of tyrosine hydroxylase (TH), a rate-limiting enzyme in the synthesis of catecholamines, and β3AR in BAT were similar between DsbA-L$^{CD4-KO}$ mice and control littermates after HFD feeding for 5 weeks (Fig. 6d), suggesting that DsbA-L in T cells may regulate thermogenesis by targeting

signaling components downstream of β3AR. Consistent with this, IFN-γ treatment significantly suppressed the β3AR agonist CL316,243-induced thermogenic gene expression in primary brown adipocytes (Fig. 6e). Based on the finding that PKA activation is critical for UCP1 expression in BAT[27], we checked

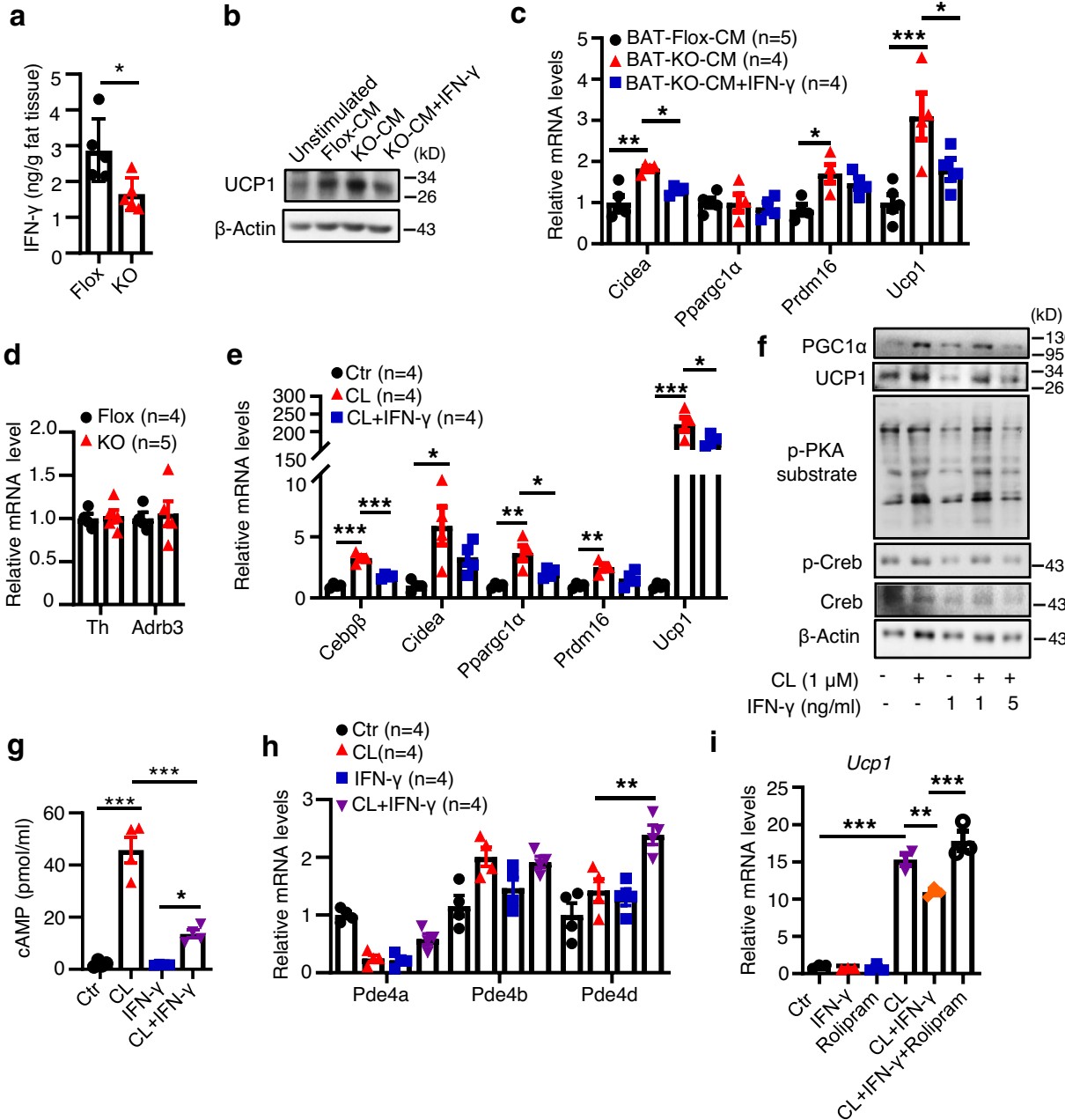

**Fig. 6 IFN-γ inhibits thermogenic gene expression in brown adipocytes. a** BAT explant cultures from DsbA-L$^{CD4-KO}$ mice and control littermates fed an HFD for 5 weeks were stimulated with a submaximal dose of PMA/ionomycin for 24 h. IFN-γ secreted into the medium was collected and quantified by ELISA. (*n* = 5/group). **b, c** Differentiated primary brown adipocytes were left untreated or treated with conditioned medium (CM) of BAT collected in **a** in the presence or absence of IFN-γ for 24 h. UCP1 expression and thermogenesis markers were tested by western blot (**b**) and qPCR analyses (**c**), respectively. (*n* = 4–5/group). **d** The mRNA levels of TH and β3AR (*Adrb3*) in the BAT of DsbA-L$^{CD4-KO}$ mice (*n* = 5) and control littermates (*n* = 4) fed an HFD for 5 weeks. **e** Differentiated primary brown adipocytes were treated with CL316,243 (1 μM) in the presence or absence of IFN-γ (5 ng/ml) for 24 h. The mRNA levels of thermogenic markers were determined by qPCR analyses. (*n* = 4/group). **f** Differentiated primary brown adipocytes were treated with CL316,243 (1 μM) in the presence or absence of IFN-γ (1 or 5 ng/ml) for 24 h. The phosphorylation and protein level of CREB, phosphorylation of PKA substrates, and protein levels of PGC1α, and UCP1 were analyzed by western blot. **g** cAMP levels accumulated in differentiated BFC adipocytes that were untreated or treated with CL316,243 (1 μM) in the presence or absence of IFN-γ (5 ng/ml) for 24 h. (*n* = 4/group). **h** mRNA levels of PDE4 family members in differentiated BFC adipocytes treated as in **g**. (*n* = 4/group). **i** Differentiated BFC adipocytes were left untreated or pre-treated with rolipram (1 μM) for 30 min, followed by stimulated with CL316,243 (1 μM) in the presence or absence of recombinant IFN-γ protein (5 ng/ml) for 24 h. Ucp1 mRNA level was determined by qPCR analyses. (*n* = 3/group). Data shown are representative of three independent experiments. All data are presented as mean ± SEM. Statistical values *p* < 0.05 (*), *p* < 0.01 (**), *p* < 0.001 (***) are determined by two-tailed unpaired Student's *t*-test (**a**, **d**) or ANOVA adjusted for multiple comparisons (**c**, **e**, **g**–**i**). Source data are provided as a Source Data File.

PKA signaling in primary brown adipocytes treated with or without IFN-γ. IFN-γ treatment greatly suppressed the phosphorylation of PKA substrates and cAMP-response element-binding protein (CREB) at Ser[133], a direct PKA-mediated phosphorylation site[28], in a dose-dependent manner (Fig. 6f). In agreement with the finding that IFN-γ suppresses cAMP accumulation in microglia and astrocytes[29], we found that IFN-γ treatment significantly decreased CL316, 243-induced cAMP levels in differentiated brown adipocytes[30] (Fig. 6g). Moreover, IFN-γ treatment markedly increased the mRNA expression of phosphodiesterase-4D (PDE4D), a predominant phosphodiesterase found in adipose tissue[31], but not the mRNA expression of PDE4B or PDE4A in the CL316,243-treated cells (Fig. 6h). To determine whether IFN-γ suppresses cAMP levels by activating PDE4D, we treated differentiated brown adipocytes with IFN-γ in the presence or absence of rolipram, a selective inhibitor of PDE4. Rolipram treatment reversed the inhibitory effect of IFN-γ on thermogenic gene expression (Fig. 6i), further confirming the role of PDE4D in IFN-γ-induced inhibition of thermogenic gene expression.

**Administration of IFN-γ reverses the enhanced BAT thermogenesis in HFD-fed DsbA-L^CD4-KO mice.** Due to technical constraints, we were unable to upregulate IFN-γ production only in T cells of the DsbA-L^CD4-KO mice. It is well known that Th1 cells and CD8^+ T cells are the main contributors of IFN-γ production in vivo[32], therefore, to determine the in vivo role of T cell-produced IFN-γ in whole-body energy expenditure, we injected recombinant IFN-γ protein into wild-type mice fed an HFD for 5 weeks as illustrated (Fig. 7a). IFN-γ injection had no obvious effect on body weight, fat mass composition, food intake, and physical activity (Supplementary Fig. 8a–c). However, by ANCOVA analyses, we found oxygen consumption was significantly reduced in IFN-γ-treated mice compared to vehicle-treated mice (Fig. 7b–d), indicating an inhibitory role of IFN-γ in energy expenditure in vivo. To determine whether the enhanced BAT thermogenesis in DsbA-L^CD4-KO mice could be reversed by IFN-γ, we injected IFN-γ into DsbA-L^CD4-KO mice and control littermates fed an HFD for 5 weeks. IFN-γ injection significantly suppressed OCR in both the floxed control mice and the DsbA-L^CD4-KO mice (Fig. 7e). IFN-γ injection also significantly reduced the protein levels of UCP1 and other thermogenic gene expression in the BAT of DsbA-L^CD4-KO mice (Fig. 7f, g). Consistently, treating mice with recombinant IFN-γ increased PDE4D mRNA levels and subsequently reduced cAMP levels in BAT of DsbA-L^CD4-KO mice (Fig. 7h, i). Taken together, these results further demonstrate the critical role of IFN-γ in the negative regulation of BAT thermogenic function.

## Discussion
The major function of adipose tissue is its inherent capacity to act as energy storage and/or energy dissipation site. Adipose tissue is heterogeneous and its function is dynamically regulated by communications between adipocytes and other cell types within the tissue, such as T cells[33]. Cumulative evidence suggests that adipose-resident CD4^+ and CD8^+ T cells are involved in over nutrition-induced inflammation[34–38]. However, the roles and mechanisms of T cells in regulating energy expenditure remain largely unclear. In the current study, we have identified DsbA-L as a critical regulator of T cell mitochondrial function and its deficiency modulates the role of T cells in the regulation of BAT thermogenesis. T cell-specific knockout of DsbA-L in mice decreased IFN-γ-producing Th1 and CD8^+ T cells in BAT, leading to decreased PDE4D expression, elevated cAMP/PKA signaling, and consequently increased UCP1 expression (Fig. 7j).

Our study identifies T cells as a key regulator of BAT function and provides insights into the mechanism by which BAT regulates thermogenesis in response to environmental changes.

BAT has been shown to contain a unique immunological compartment that is important for physiologic responses to thermogenesis[39,40]. However, the mechanisms regulating the crosstalk between immune cells and adipocytes in BAT remain largely unclear. In the current study, we showed that UCP1 expression in BAT was inhibited by T cell-produced IFN-γ, which suppressed the cAMP-PKA signaling pathway via a PDE4D-dependent mechanism (Fig. 6b, c, g, h). Our results are consistent with the previous finding that repression of IFN-γ signaling contributes to the browning of adipose tissue[10,41]. We also showed that IFN-γ administration reversed the enhanced OCR in BAT of DsbA-L^CD4-KO mice (Fig.7e). However, the exact mechanism by which DsbA-L deficiency downregulates IFN-γ levels in T cells remains unknown. It is reported that inhibition of the electron transport chain complex significantly reduces IFN-γ production[42]. Given that DsbA-L deficiency dysregulated mitochondrial function (Fig. 2), it is thus possible that DsbA-L deficiency reduces IFN-γ production in T cells via a mitochondria-dependent mechanism. We also found that DsbA-L deficiency increased Treg cell accumulation in the BAT of HFD-fed mice (Fig. 5e). Treg cells have recently been shown to regulate diet-induced BAT thermogenesis upon short-term HFD feeding[11]. Thus, it is also possible that DsbA-L deficiency in T cells may stimulate diet-induced BAT adaptive thermogenesis and energy expenditure by inhibiting IFN-γ production and by promoting Treg cell accumulation. While the precise mechanism by which DsbA-L deficiency increases Treg expansion remains unknown, there is some evidence showing that IFN-γ negatively regulates Treg differentiation and accumulation[43–45]. Thus, DsbA-L deficiency may increase Treg cells by reducing IFN-γ levels. Further studies are necessary to validate these possibilities.

The term "diet-induced thermogenesis", which is used rather vaguely, is initially used to describe marked but transient increase in post-prandial energy dissipation, which comprises of energy required for the processing, transport, and storage of nutrients from the meal and a facultative component of heat energy[46,47]. Given that T cell DsbA-L deficiency affects HFD- but not ND-induced thermogenic gene expression and energy expenditure, we postulate that T cells regulate a diet-induced adaptive thermogenic response, or the so-called "diet-adaptation-recruited norepinephrine-induced thermogenesis"[3,48,49], rather than the acute diet-induced thermogenic responses. The immune system has been proposed to be tightly associated with DIT from an evolutionary perspective and the immune-regulation of DIT leads to better host survival outcomes potentially by limiting the otherwise uncontrolled expansion of the Firmicutes population[50]. Consistent with this view, a recent study has showed that Treg cells, which are efficient in suppressing effector T cell immune function, were greatly induced by β3AR signaling following short-term HFD feeding and contributed to DIT[11]. In our study, we found that decreased T cell immune responses, manifested by decreased IFN-γ production, promoted diet-induced BAT adaptive thermogenesis in DsbA-L^CD4-KO mice, further supporting the concept that compromised T cell immune function could facilitate DIT.

In this study, we observed that T cell-specific knockout of DsbA-L promoted oxygen consumption and energy expenditure in HFD-fed mice under both room temperatures and thermoneutral conditions (Fig. 4j, k) but had no significant effect on cold-induced thermogenic gene expression (Supplementary Fig. 4a–c). A possible explanation for the specific effect of DsbA-L on DIT but not CIT maybe that cold exposure significantly

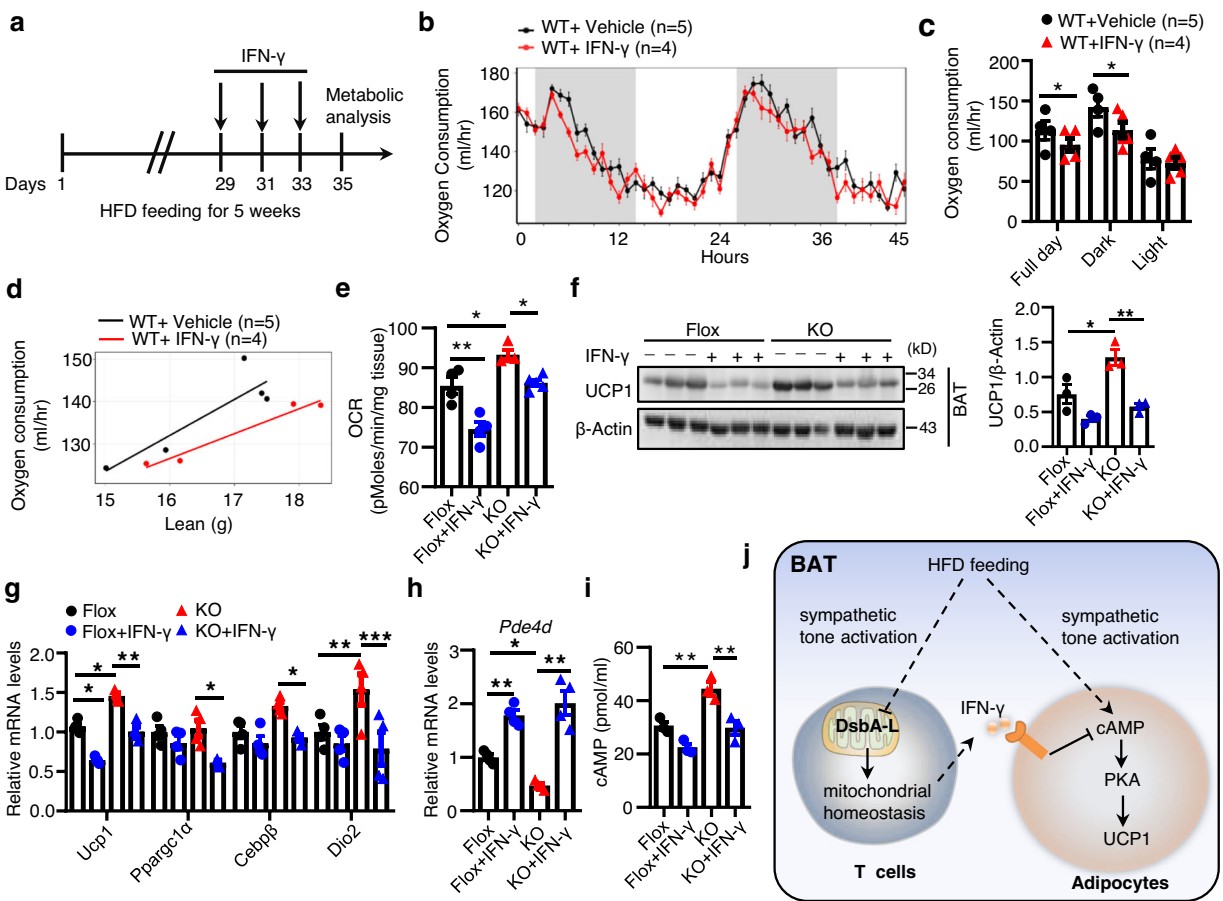

**Fig. 7 Administration of IFN-γ reverses the enhanced BAT thermogenesis in HFD-Fed DsbA-L$^{CD4-KO}$ mice. a** Schematic for IFN-γ dosing and metabolic analyses IFN-γ (100 μg/kg) was injected intraperitoneally every other day for the last week. **b** The oxygen consumption of wild-type mice fed an HFD for 5 weeks with (n = 4) or without IFN-γ (n = 5) administration for the last week. **c** The average oxygen consumption shown in **b**. **d** ANCOVA analyses of average VO$_2$ (ml/h) in **c** by lean mass were shown. **e** The basal OCR of BAT collected from DsbA-L$^{CD4-KO}$ mice and control littermates fed an HFD for 5 weeks with (n = 4–5/group) or without IFN-γ (n = 4/group) treatment for the last week. **f** Western blot analyses of PGC1α and UCP1 levels in BAT of DsbA-L$^{CD4-KO}$ mice and control littermates fed an HFD for 5 weeks with or without IFN-γ treatment for the last week. (n = 3/group). **g** mRNA levels of thermogenic genes in BAT of DsbA-L$^{CD4-KO}$ mice and control littermates fed an HFD for 5 weeks with or without IFN-γ treatment for the last week. (n = 4/group). **h** The mRNA levels of PDE4D in BAT of DsbA-L$^{CD4-KO}$ mice and control littermates fed an HFD for 5 weeks with or without IFN-γ treatment for the last week. (n = 4/group). **i** cAMP levels accumulated in BAT of DsbA-L$^{CD4-KO}$ mice and control littermates fed an HFD for 5 weeks with or without IFN-γ treatment for the last week. (n = 3/group). **j** A proposed model of the mechanism by which T cell DsbA-L regulates energy expenditure in BAT. A diet-induced signal stimulates the sympathetic nerve system and downstream cAMP-PKA activation to promote UCP1 expression in BAT. Meanwhile, HFD feeding leads to decreased DsbA-L expression in BAT-resident T cells. DsbA-L deficiency blocks T cell mitochondrial function and decreases IFN-γ-producing Th1 and CD8$^+$ T cells accumulation in BAT. IFN-γ could act on brown adipocytes and inhibit UCP1 expression and cAMP accumulation via promoting PDE4D expression. Decreased IFN-γ production in BAT thus contributes to enhanced diet-induced BAT thermogenic capacity and alleviating over obesity. Data shown are representative of three independent experiments. All data are presented as mean ± SEM. Statistical values $p < 0.05$ (*), $p < 0.01$ (**), $p < 0.001$ (***) are determined by ANOVA adjusted for multiple comparisons (**e**, **f** (right panel), **g-i**) or ANCOVA test (**c**). Source data are provided as a Source Data File.

decreased IFN-γ expression in BAT of control mice, which is consistent with previous studies showing that cold exposure suppresses immune responses in adipose tissue[51–53]. However, cold exposure did not further inhibit IFN-γ expression in BAT of DsbA-L$^{CD4-KO}$ mice (Supplementary Fig. 6h), probably because IFN-γ expression was already reduced to its basal level in BAT of the mice. The similarly low levels of IFN-γ in the BAT of cold-exposed control mice and DsbA-L$^{CD4-KO}$ mice may explain why cold exposure-induced thermogenesis was comparable between DsbA-L$^{CD4-KO}$ mice and control littermates. Another possible explanation may be that the stimulatory effect of cold on BAT thermogenesis is much more pronounced than HFD feeding. Thus, it may mask the promoting effects of IFN-γ reduction on thermogenesis in DsbA-L$^{CD4-KO}$ mice.

We found that the serum levels of IFN-γ were similar between HFD-fed DsbA-L$^{CD4-KO}$ mice and control littermates (Fig. 5h), though DsbA-L$^{CD4-KO}$ mice showed an enhanced energy expenditure (Fig. 4e). Unlike acute infectious disease-induced strong systemic inflammation, which causes a significant increase in serum IFN-γ levels[54], HFD feeding has been shown to induce a low-grade inflammation state in mice that has little effect on serum IFN-γ levels[55–57]. In addition, BAT-resident T cells account for only a small proportion of T cells in the whole body, and other IFN-γ-producing cells, such as NK cells, could also contribute to the serum IFN-γ levels. We believe that inflammation in BAT of HFD-fed mice is of low-grade and tissue-restricted, which will affect the thermogenic function of BAT but do not cause strong inflammatory changes throughout the body.

In summary, we have identified T cell-specific DsbA-L as a critical regulator of T cell mitochondrial function and whole-body energy homeostasis. T cell-specific knockout of DsbA-L markedly decreases IFN-γ production but promotes Treg accumulation, thus facilitating diet-induced BAT thermogenesis and alleviating obesity. Our study proposes a critical mechanism underlying the immune-regulation of metabolic homeostasis and suggests that targeting adipose tissue-resident T cells may have therapeutic potential for the treatment of diet-induced obesity and its associated metabolic diseases.

## Methods

**Mice and cell line.** T cell-specific DsbA-L knockout mice (DsbA-L[CD4-KO]) were generated by crossing DsbA-L floxed mice (*DsbA-L[f/f]*)[17] with CD4-Cre mice (Jackson Laboratory; Cat No. 017336). DsbA-L[CD4-KO] mice lack DsbA-L in both CD4[+] and CD8[+] T cells due to the DsbA-L gene deletion in the CD4[+]CD8[+] double-positive stage during thymic maturation. CD3[+] T cells are mainly composed of CD4[+] T cells and CD8[+] T cells, so DsbA-L deletion can also be detected in isolated CD3[+] T cells. Mice were kept in a specific pathogen-free animal facility at (23 ± 1)°C, 50–60% relative humidity, and a 12-h light/dark cycle. Mice had free access to food and water and fed ad libitum either an ND or an HFD (60% fat; Research Diets, New Brunswick, NJ). All animal studies were performed under a protocol approved by the Central South University Animal Care and Use Committee and in compliance with all relevant ethical regulations for animal testing and research. The brown adipocyte cell line BFC was a generous gift from Dr. Jiandie Lin (Univ. of Michigan)[30].

**IFN-γ secretion from adipose tissue ex vivo.** Explant cultures of adipose tissue were performed as described previously[58]. Briefly, eWAT, iWAT, and brown adipose tissue (BAT) were dissected from HFD-fed DsbA-L[CD4-KO] mice and their control littermates. Adipose tissues (about 100 mg) were then weighed, minced, and placed into 6-well tissue culture dishes with either T cell activation medium (1640 medium containing 10% FBS, L-Glutamine (1 mM), β-Mercaptoethanol (50 μM)), and a submaximal dose of PMA (5 ng/ml), and ionomycin (75 ng/ml). Conditioned medium (CM) was collected 24 h post-treatment. IFNγ levels were determined using the ELISA kit (Biolegend, San Diego, USA).

**Tissue homogenization protocol for ELISA.** Tissue homogenization experiments were performed according to the protocol as described[10]. Briefly, adipose tissue (0.05 g) was homogenized in 200 μl cold HBSS supplemented with proteinase inhibitors. After centrifugation at 400×*g* for 15 min at 4 °C, IFN-γ levels in the supernatant of the tissue homogenates were measured by ELISA.

**Cold exposure studies.** Male DsbA-L[CD4-KO] mice and their floxed littermates (8-week old) were individually or doubly housed at 6 °C in a non-bedded cage with access to food and water. At the end of the experiment, mice were sacrificed and fat tissue was isolated for gene and protein expression analyses.

**Glucose tolerance test (GTT) and insulin tolerance test (ITT).** Male DsbA-L[CD4-KO] mice and their floxed littermates at 8 weeks of age were fed a ND or a 60% HFD (Research Diets Inc; USA) for 12 weeks. For glucose tolerance tests (GTTs), mice were fasted overnight and challenged with an intraperitoneal injection of glucose (2 g/kg). For insulin tolerance tests (ITTs), mice were fasted for 6 h, followed by an intraperitoneal injection of insulin (0.75 U/kg). Blood glucose levels were monitored using the ACCU-CHEK active glucometer (Roche).

**Flow cytometry analysis and ELISA.** Mouse SVFs were firstly isolated as previously described[59]. Briefly, adipose tissue was carefully excised, minced, and digested with 1.5 g/l type 2 collagenase (Sigma-Aldrich) for 25 min at 37 °C, with shaking. Digested cells were filtered with a 100 μm nylon screen, washed, and centrifuged for 5 min to pellet the SVFs from floating mature adipocytes.

To detect the expression of surface molecules, splenocytes and SVFs were first incubated with an anti-Fc receptor (Biolegend, San Diego, CA) to reduce nonspecific binding of antibodies, followed by incubation with the indicated antibodies for 20–30 min at 4 °C. For analysis of transcription factor expression, surface labeled cells were fixed, permeabilized, and stained with indicated antibodies for 40–50 min at 4 °C with Transcription Factor Buffer Set (BD Biosciences, San Jose, CA) according to the manufacturer's instructions. For analysis of intracellular IFN-γ, cells were stimulated with PMA (50 ng/ml; Beyotime, Shanghai, China) and ionomycin (750 ng/ml; Millipore, Darmstadt, Germany) for 6 h with the addition of Brefeldin A (10 μg/ml; Beyotime, Shanghai, China). Cells were harvested, washed, fixed, permeabilized with the Fixation/Permeabilization Solution Kit (BD Biosciences, San Jose, USA), and stained with the APC-IFN-γ antibody. Appropriate fluorescein-conjugated, isotype-matched, irrelevant mAbs were used as negative controls. Cells were collected on a BD Canto II using the BD FACS Diva software (BD Biosciences, San Jose, USA). Commercial

antibodies used in this study including FVS520, APC-Cy7-CD45, Percp-CD4, APC-CD62L, PE-Cy7-CD44, Alexa fluor 647-GATA3, FITC-γδ TCR (BD Biosciences, San Jose, USA), and Zombie NIR, PE-Cy7-CD45, PE-CD3, PE-CD4, PE-Cy7-CD8, PE-CD8, PE-CD25, APC-CD8, APC-IFN-γ, Alexa fluor 647-Foxp3, Percp-Cy5.5-CD11b, APC-F4/80, FITC-CD206 (Biolegend, San Diego, USA), and PE-Siglec-F, Alexa fluor 647-Foxp3, FITC-γδTCR (ebioscience, CA, USA). All antibodies were diluted according to the manual from the manufacturer's website. Dead cells and doublets were removed by dead-cell dye staining (FVS520 or Zombie NIR) and FSC-A/FSC-H gating, respectively. Data were analyzed by FlowJo V10 (BD Biosciences, San Jose, USA).

IFN-γ and IL-4 protein levels were determined using IFN-γ and IL-4 ELISA MAX Deluxe Sets (Biolegend, San Diego, CA), respectively.

**Mitochondrial function assays.** For mitochondrial mass, mitochondrial membrane potential, and mitochondrial ROS measurements, mouse SVFs collected from BAT and iWAT Tregs were incubated with 100 nM Mito-Tracker Green, 100 nM Mito-Tracker Red, and 5 uM mitoSOX (Life Technologies, Carlsbad, USA), respectively, at 37 °C for 15–30 min and then analyzed by flow cytometry. A combination of Mito-Tracker Green (mitochondrial membrane potential-independent) with Mito-Tracker Red (mitochondrial membrane potential-dependent) was used to distinguish between respiring mitochondria and dysfunctional mitochondria[20].

OCR and ECAR were measured using the XF24 extracellular flux analyzer (Seahorse Bioscience, Billerica, USA), as described[60]. Approximately $5 \times 10^5$ T cells per well were used for seahorse analyses. T cells were attached to XF24 plates using Cell-Tak (Corning, Bedford, USA) and measured in XF media (non-buffered RPMI 1640 containing 25 mM glucose, 2 mM L-glutamine, and 2 mM sodium pyruvate). Basal mitochondrial respiration was measured in untreated cells. The cells were further treated with 0.25 μM oligomycin, 1.5 μM carbonyl cyanide trifluoromethoxy phenylhydrazone (FCCP), and 0.1 μM rotenone, and 1 μM antimycin A. All of the reagents were from Sigma-Aldrich, USA. Results of OCR and ECAR were normalized to protein content. Each measurement was repeated 4–6 times to achieve consistent and reliable results.

For tissue measurement, studies were carried out as described[61]. Briefly, adipose tissues were collected from HFD-fed DsbA-L[CD4-KO] and their floxed littermates. Tissues were rinsed with unbuffered KHB medium containing 111 mM NaCl, 4.7 mM KCl, 2 mM MgSO₄, 1.2 mM Na₂HPO₄, 0.5 mM carnitine, and 2.5 mM glucose, and then cut into pieces (~8 mg). After extensive rinsing, each piece of the tissue was placed in a single well of the Islet Flux plate (Seahorse Bioscience, Billerica, USA) and covered with a customized screen that allows for free perfusion while minimizing tissue movement, followed by the addition of 500 μl of KHB buffer. The basal OCR of each animal was measured with three independent pieces and results were normalized to tissue weight.

**Mitochondrial calcium measurement.** T cells grown on cell culture dishes coated with Cell-Tak (Corning, Bedford, USA) were loaded with 2.5 μM mitochondrial calcium indicator Rhod2 (Invitrogen, Carlsbad, USA) for 30 min at 37 °C. Dye-loaded T cells were rinsed with PBS (containing 2% FBS) twice, and then incubated with hamster anti-mouse CD3 antibody (Biosciences, San Jose, USA) at room temperature for 30 min. Rhod2 baseline fluorescence was measured by Zeiss lsm780 confocal microscope system (Zeiss, Oberkochen, Germany) and cell activation was initiated by cross-linking with the secondary anti-hamster antibody (Biolegend, San Diego, USA). Rhod2 fluorescence has been acquired at 568 nm excitation for up to 10 min.

**Quantitative real-time PCR (qPCR).** Mouse tissues were snap-frozen in liquid nitrogen and stored at −80 °C until use. Total RNAs were extracted using the TRIzol Reagent (Life Technologies, Carlsbad, USA) following the manufacturer's instructions. mRNAs were reverse-transcribed and amplified using a 7900HT Real-Time PCR System (Applied Biosystem, USA). The primer sequences for the genes are shown in Supplementary Table 1.

**Western blots.** Western blots were performed using antibodies to p-PKA substrate, p-CREB, total-CREB, p-DRP1, total-DRP1, p-GSK3β, total-GSK3β (Cell Signaling, Danvers, USA), UCP1, ERK1/2, Actin (Sigma, St. Louis, MO), PGC1α (Abcam, Cambridge, UK), OPA1, MFN2 (Proteintech, Rosemont, USA), and home-made antibodies against DsbA-L. Quantification of Western blot results was conducted using NIH Image J software (National Institutes of Health, Bethesda, Md, USA).

**Isolation of T cells.** CD3[+] and CD8[+] T cells were isolated from the spleen by staining with PE-CD3 antibody (Biolegend, San Diego, USA) and PE-CD8 antibody (Biolegend, San Diego, USA), respectively, for 20 min in the dark in a refrigerator. Cells were washed and then incubated with anti-PE microbeads (Miltenyi, Auburn, USA). After washing, CD3[+] and CD8[+] T cells were finally isolated by using the magnetic beads separation system (Miltenyi, Auburn, USA). CD4[+] T cells were also isolated from the spleen by magnetic beads separation system using a mouse CD4[+] T cell isolation kit (Miltenyi, Auburn, USA) as previously described[62].

**H&E and immunohistochemical staining**. H&E and immunohistochemical staining were performed as described[59].

**Bodyweight, body composition, food intake, and gut absorption measurement**. Mouse body weight was monitored weekly. Mouse body composition was determined by using the Minispec Body Composition Analyzer LF50 (Bruker, Germany). Food intake and feces production were measured from individually housed mouse every day for 4 days. The feces were dried and ground to a fine powder before subjecting them to an oxygen bomb calorimeter (IKA, Germany) according to the manufacturer's instructions. Calorie excretion was calculated by multiplying the produced feces with the calories content per gram of feces.

**Comprehensive lab animal monitoring system**. Indirect calorimetry experiments were performed with a Comprehensive Lab Animal Monitoring System (CLAMS, Columbus Instruments, USA). Mice were fed ad libitum and housed in the CLAMS and maintained at either sub-thermoneutrality (22 °C) or thermoneutrality (30 °C) conditions for 3 days, respectively. For each experimental condition, metabolic variables were adjusted for differences in body composition by ANCOVA in the R programming language with a custom package for indirect calorimetry experiments (CalR).

**IFN-γ administration**. For IFN-γ-induced suppressing effects on BAT adaptive thermogenesis, DsbA-L$^{CD4-KO}$ mice and control littermates were fed an HFD for 4 weeks before intraperitoneal injection with PBS (vehicle) or IFN-γ (100 μg/kg, PeproTech, Cranbury, USA) every other day for the 5th week. Mice were collected after a total of 5 weeks of HFD feeding.

**Mitochondrial dynamics**. T cells adhered to microscopic imaging chambers coated with cell-tak (Corning, Bedford, USA). Cells were stained for 30 min at 37 °C with 100 nM Mito-Tracker Green (Beyotime, Shanghai, China) and 10 μg/ml Hoechst 33342 (Beyotime, Shanghai, China) for nuclear staining. For imaging analysis, cells were rinsed with DPBS and resuspended. Mitochondrial morphology was determined by confocal microscopy (Zeiss, Oberkochen, Germany) immediately. Images of several fields were taken for each treatment group.

**Statistics**. Independent experiments were repeated at least three times, and the data are presented as mean ± SEM. Unless otherwise indicated in the figure legends, statistical significance was determined with unpaired, two-tailed Student's t-test or ANOVA test for multiple comparisons with Tukey's test for post-hoc corrections. The Analysis of Covariance (ANCOVA) test was performed to analyze differences in oxygen consumption (VO$_2$) between the control and experimental groups while statistically controlling for the effects of covariate lean mass. Statistical significance is indicated as ns, not significant, *$p < 0.05$, **$p < 0.01$, ***$p < 0.001$. All statistical analysis was performed using the Prism 8.0 software (Graphpad, San Diego, CA).

**Reporting summary**. Further information on research design is available in the Nature Research Reporting Summary linked to this article.

## Data availability

The data that support the findings of this study are available from the corresponding author upon reasonable request. Source data are provided with this paper.

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

## Acknowledgements

This work was supported by grants from the National Natural Science Foundation of China (Grant Nos. 81730022 and 81600671), and an Innovative Basic Science Award from the American Diabetes Association (1-19-IBS-147), and grants from the National Key R&D Program of China (2018YFC2000100, 2019YFA0801903), and grants from the Natural Science Foundation of Hunan Province, China (Grant No. 2019JJ50867).

## Author contributions

H.Z. and X.P. designed experiments and analyzed data. H.Z., X.P., J.H., L.W., H.L., J.Z., Y.Z., G.L., and Y.J. performed experimental work. J.B., M.L., J.J.Z., and Z.Z. contributed to discussion and manuscript editing. H.Z. and F.L. conceived of, supervised the project, and wrote the manuscript. All authors discussed and approved the manuscript.

## Competing interests

The authors declare no competing interests.
