## [Peer Review File · Nature Communications]

Reviewers' Comments:

Reviewer #1:

Remarks to the Author:

In the present manuscript the authors present evidence that DsbA-L CD4 KO mice fed a HFD have higher energy expenditure at room temperature and under thermoneutral conditions. Upon hypercaloric challenge, DsbA-L CD4KO mice present with improved metabolic indices, reduced body weight gain and enhanced BAT thermogenesis.

The authors identify IFN γ as critical mediator that limits the upregulation of thermogenic genes as response to diet-induced thermogenesis (DIT). DsbA-L CD4KO BAT harbor reduced IFN γ levels and the authors conclude that this IFN γ reduction leads to the improved overall metabolic indices upon hypercaloric challenge. The authors confirm the critical role of IFN γ in suppressing the upregulation of thermogenic genes in adipocytes as response to beta3-adrenergic stimulation (e.g. CL treatment).

The data of this manuscript support the critical role of adipose tissue-residing immune cells in modulating tissue function in response to external stimuli such as cold or hypercaloric feeding. These findings are of interest, however, in the view of this reviewer several concerns arise that should be addressed to strengthen the conclusions:

The authors state that the DsbA-L KO affects mitochondrial biogenesis/function in T cells. It is known that different T cell subsets utilize different substrates to fuel their metabolism. Beta-oxidation in the mitochondria is one important pathway. While the authors show that the OCR is reduced in DsbA-L CD4 KO cells, ECAR is not affected (and not shown). KO T cells present with reduced ATP production and reduced mitochondrial content. Therefore, it is of superior relevance to understand how this KO affects the different adipose tissue T cell subphenotypes since their capability to deal with this KO might be substantially different – thereby affecting the whole read out.

Specific comments to the figures:

Figure 1: Does not fit on one page.

For all Mitotracker/mitSOX experiments: since no individual points are plotted, N numbers have to be provided in the figure legend. Please revise.

E-F) Example of the gating strategy for quantification of % mitoSOX positive cells?

Figure 2:

B) How many T cells were used for Seahorse analyses? Not stated in methods section and also not in the literature cited.

Figure 3:

B) KO mice present with reduced BW C) total fat mass is reduced in KO mice.  E) tissue weight of BAT, iWAT, pWAT reduced – also when the authors normalize it to the reduced BW?

H+I) GTT – Please include the AUC for GTT and ITT

Figure 4 – does not fit on one page.

A) Quality of western blot. Beta-actin as reference protein suggests that different amounts of protein were loaded on the gel. Quantification of this blot is questionable. Nevertheless, it seems that KO mice have higher UCP1 content in BAT.

D) Oxygen consumption – which group is shown? HFD or ND group?

G) which tissue is shown?

I) Quality of Western blot should be optimized.

Figure 5 – does not fit on one page.—see also related comments in Figure S5.

IFN γ levels are significantly reduced only in BAT in KO animals (B), but there are less IFN γ + CD4+ and also CD8+ T cells observed in iWAT (D-G). Here, not only the frequency but also total numbers of IFN γ + T cells are reduced in iWAT, without affecting total IFN γ protein in tissue lysates

(B). Please comment. The authors should also include eWAT in their analysis in B, since eWAT was also analyzed in D-G with significant differences and is a relevant adipose tissue depot affecting systemic metabolism.

A) Please stick to nomenclature for gene expression. IL-2  IL2

B) Y axis – pg/ml; is it rather pg/g adipose tissue? Did the authors determine the amount of adipose tissue analyzed here volumetrically?

Figure 6

A) Y axis is now pg/ml/g fat tissue – see comment for Fig 4B. What is correct?

E) Is the quantification of the WB correct? Focusing on UCP1 expression in KO cells treated with IFN γ : first replicate shows nearly no UCP1 protein but strong beta-actin signal. Normalizing that, this leads to an UCP1/beta actin ration <1. In comparison to the second replicate, the related summary graph needs clarification with respect to the size of the error bar and the absolute value.

Supplemental Figures

Figure S1

The authors should include the analysis of Treg frequencies to test whether thymic Treg development is impaired. The critical role of Tregs in maintaining tissue homeostasis and the response to cold was given in the introduction.

A) Quality of WB: Why is there no difference in CsbA-L protein expression in CD3+ T cells from floxed vs KO? It should look the same as for CD4+ and CD8+ T cells. Moreover, it should be visible in spleen (or at least a trend) since the majority of cells in the spleen are CD4+ and CD8+ T cells. Please comment. (starting text line 130)

B) Please adhere to the nomenclature for mRNAs

Figure S5

The authors should include an analysis of – at least – Foxp3+ Treg frequencies here (and earlier) since Tregs are known to be critically involved in regulating thermogenesis, limiting inflammation, maintaining tissue homeostasis. The fact that CD4+ frequencies are not altered between KO and floxed animals in the presence of altered IFN γ levels just indicates that there is likely a shift between T cell subpopulations going on. This altered immune cell composition significantly contributes to the overall metabolic effects observed and therefore, should be analyzed carefully.

Further comments:

Additionally to the WB in Figure S1, to ensure that the DsbA-L Knockout is restricted to the T cell lineage, the authors should provide a gel showing that no ectopic recombination has occurred in e.g. adipose tissues, the liver and muscles. This can be done easily by standard PCR, detecting and amplifying the recombined allele. Just to make sure that the observed effects are T cell-mediated and not due to Cre leakiness affecting major metabolic tissues.

line 160 – The DsbA-L CD4 KO had no effect on eWAT volume or adipocyte size, but presented with reduced inflammatory gene expression (Fig 3D-G and Fig S2E). In Figure S2E there is a significant reduction in Cd3e mRNA levels in eWAT, indicating that T cell levels might be affected in the tissue upon T cell-specific DsbA-L KO. This raises the question, which T cell population is affected and whether regulatory T cell frequencies are reduced. Detailed analyses should be included.

After hypercaloric challenge (12 weeks), DsbA-L CD4 KO mice present with an increase in thermogenesis in BAT. Whole body oxygen consumption and energy expenditure are increased in KO mice. Mitochondrial content and genes related to FAO are increased in BAT of KO mice. How does the long-term hypercaloric challenge affect adipose-resident T cells in KO vs floxed mice? The authors should include detailed analyses here, since the observed effects on whole body metabolism are huge keeping in mind that the authors are analyzing a T cell specific KO and therefore, the effects observed should be a result of the altered T cell population/frequency/functionality upon DsbA-L KO. The afterwards performed experiments with 5 wk HFD and IFN γ injection are not sufficient in order to reveal the underlying mechanism by which a T cell specific DsbA-L KO affects systemic metabolism in hypercaloric conditions.

"Data not shown" – e.g. ECAR of T cells (line 134); no difference in BW, fat mass, food intake or physical activity upon IFN γ injections (line 315); line 324; line 395 no difference in IFN γ expression in BAT T cells upon cold exposure.

Line 424 – please define the exact composition of the T cell activation medium and especially define the submaximal doses of PMA and ionomycin. Was the media supplemented with pyruvate, NEA, etc.?

The authors should provide a full gating strategy in the supplement, as they did it in Fig. 4H. Otherwise, it is difficult to evaluate flow cytometric analyses.

Did the authors use a dead cell stain? This is especially important when analyzing immune cells isolated from adipose tissues – since the isolation protocol itself, as well as the lipid rich environment during isolation easily kills immune cells.

As depicted in Fig. 4H, it seems that no doublet exclusion (and no live/dead cell discrimination, as mentioned above) was performed, which is critical to eliminate readout errors.

Related to the methods section and specifically the isolation of T cells – CD3 $^+$ and CD8 $^+$ T cells were isolated using PE-labeled antibodies and anti-PE beads, which is fine. Then the isolated population contains both, CD4 $^+$ and CD8 $^+$ T cells. While CD4 $^+$ T cells were isolated separately, but not first, as it seems. For which experiments have these enriched cells been used?

Line 547 – which post-hoc test was used to adjust for multiple comparison? Please clarify.

The authors should carefully revise their figure legends with regard to

1. Include N numbers for all bar graphs
2. Check for spelling mistakes (some are listed below) and please, adhere to nomenclature

Text related

Line 53 – relied (rely)

Abbreviation Treg not introduced

Line 102 – Is T cell mitochondrial mass really a biomarker for T cell biogenesis? Consider revising

Line 106 – m ice  mice

Line 107 – to a lessER extent

Line 145 – (

Line 364 - ... by which DsbA-L deficiency increases Treg expression remains unknown.

Figure legends - °C missing (e.g. figure 4); several times group  group

Reviewer #2:

Remarks to the Author:

Comments to the manuscript: NCOMMS-20-09926

The manuscript entitled "DsbA-L deficiency in T cells promotes diet-induced thermogenesis through suppressing INF γ production" by Zhou et al, investigates the role of Disulfide bond A oxidoreductase-like (DsbA-L) protein in diet-induced thermogenesis. The authors found that DsbA-L expression in adipose tissue resident T cells is decreased when mice are fed a HFD and that this decrease correlated with a decrease in T cells mitochondrial function. DsbA-L specific deletion in T cells enhanced diet-induced thermogenesis in BAT and protected mice from diet-induced obesity, hepatosteatosis and insulin resistance. More in detail, DsbA-L deletion in T cells impaired mitochondrial function, reduced INF γ production and activated PKA by decreasing phosphodiesterase-4D expression. That leads to increased BAT thermogenesis.

The novelty of the manuscript is that the authors with this work uncovered a new mechanism by which T cells crosstalk with BAT regulate brown fat thermogenesis and whole body energy homeostasis. That highlight a therapeutic potential of targeting T cells to treat diet-induced obesity and related metabolic complications.

The manuscript is well written, rendering it easy to read. However some of the data shown do not completely support the conclusions of the authors. Some validations and further experiments suggested in the comments are needed to strength the study.

Major comments

Fig1C-D. What shown in Fig 1C-D seems to be the opposite of what is described in the text. It seems that the HFD increases MMP in CD4+ and CD8+ T cells in BAT (Fig 1C) and decreases MMP in CD4+ and CD8+ T cells in iWAT (Fig 1D).

To specifically delete DbsA-L in T cells, the authors use a CD4-Cre mouse. If the Cre mouse is CD4+ specific, how it is possible that there is deletion also in CD8+ and CD3+ cells? Can the authors comment on that?

Fig2B. DbsA-L depletion in T cells leads to a decrease in basal and maximal respiration in CD4+ cells and only a decrease in the max respiration in CD8+. Do the authors have an explanation for this difference? What happened to CD3+ cells?

FigS2E. The authors suggest that the DbsA-L T cells specific ko mouse present reduced inflammation in eWAT vs controls. In this panel, only one inflammatory gene is shown to be reduced, the authors should provide more evidence of a reduced inflammation.

Fig3H-I. The authors should normalise the GTT data to the basal and provide the AUC. Moreover, what are the values of fasting glucose and insulin of the ko mice vs control?

Fig4D-F. The authors should normalise the indirect calorimetry data by the lean mass of the mice or using an ancova test.

Fig4G-H. Is the basal OCR of iWAT increased too? The authors should provide this data. Moreover, how is the expression of *cpt1* and *ppara* in the iWAT?

Fig4J. CD137 can not really be considered anymore as a beige marker (see paper "CD137 negatively affects "browning" of white adipose tissue during cold exposure." Srivastava RK, Moliner A, Lee ES, Nickles E, Sim E, Liu C, Schwarz H, Ibáñez CF. *J Biol Chem*. 2020 Feb 14;295(7):2034-2042. doi: 10.1074/jbc.AC119.011795.).

Fig4K-M. The authors should normalise the indirect calorimetry data by the lean mass of the mice or using an ancova test. Same for Fig7B and C.

It would be interesting if the authors could show also that the BAT of the ko mice vs controls show increased max respiration capacity after norepinephrine treatment at thermoneutrality.

Fig.S4. Why the DbsA-L ko mice have increased DIT and not CIT? what is the explanation for that? Can the authors comment on it?

Fig5. The authors should provide the same data shown in this figure for BAT, iWAT, eWAT of mice exposed to cold, to show that the effect is specific for DIT.

If the total amount of serum $\text{INF}\gamma$ is unchanged in the ko mice vs controls on HFD, how is possible the effect of increased DIT is due to a decrease in $\text{INF}\gamma$ in vivo? The authors should provide an explanation for that.

Fig.7 How physiological is the amount of $\text{INF}\gamma$ given to the mice? How was the amount to be injected calculated? Is that amount similar to the one measured in the serum of wt mice after HFD challenge?

Minor comments

Line 145 of the text. The authors should remove the "(".

Reviewer #3:

Remarks to the Author:

In this study, the authors have demonstrated that in adipose-resident T cells, DsbA-L deficiency led to mitochondrial dysfunction, reduction of INF- γ production and activation of PKA by reducing phosphodiesterase-4D expression, thereby increasing BAT thermogenesis. These findings are interesting and novel, and revealed a new role for DsbA-L in obesity. Several minor concerns need to be addressed.

1. In Fig3H and I, the GTT and ITT experiments were applied to detect glucose tolerance and insulin sensitivity, whether phosphorylated protein expression (PKB and GSK3 β) can be detected to represent insulin resistance? Please refer (Diabetes 2018 04;67(4)).
2. In Fig2, the author proved that DsbA-L was a critical regulator of T cell function, whether DsbA-L deficiency affects mitochondrial dynamics in T cells?
3. In the study, the author detected the function of mitochondria, such as OCR, ATP and mtDNA. What's the potential role of calcium ions within mitochondria in these changes under the condition of DsbA-L deficiency? The authors should discuss this point.
4. In Fig4, Which subtype of CPT1 is it? CPT1A, CPT1B or CPT1C?

We were very glad to know that the reviewers considered our work novel, of interest, and highlight a therapeutic potential of targeting T cells to treat diet-induced obesity and related metabolic complications. In addition, our data “support the critical role of adipose tissue-residing immune cells in modulating tissue function in response to external stimuli”. We thank you for your suggestions and encouragement in resubmitting this manuscript. We also thank the reviewers for their insightful and constructive comments regarding our work. In accordance with your suggestions and those of the reviewers, we have carried out additional experiments and revised our manuscript (changes are highlighted by red. My responses to comments on the manuscript are as follows:

Reviewer #1:

1. The authors state that the DsbA-L KO affects mitochondrial biogenesis/function in T cells. It is known that different T cell subsets utilize different substrates to fuel their metabolism. Beta-oxidation in the mitochondria is one important pathway. While the authors show that the OCR is reduced in DsbA-L CD4 KO cells, ECAR is not affected (and not shown). KO T cells present with reduced ATP production and reduced mitochondrial content. Therefore, it is of superior relevance to understand how this KO affects the different adipose tissue T cell subphenotypes since their capability to deal with this KO might be substantially different – thereby affecting the whole read out.

Response:

We thank the reviewer for the constructive comments. ECAR results have been included in our revised manuscript (Supplementary Fig.1d). Reduced OCR but unaltered ECAR were observed in DsbA-L-deficient T cells compared with wild-type T cells, suggesting that DsbA-L in T cells are essential for oxidative phosphorylation (OXPHOS) but not glycolysis. Since adipose-resident T cells are reported to be effector memory T cells that mainly adopt OXPHOS to meet their energy demands^{1,2}, DsbA-L-deficiency may thus affect T cell metabolism in adipose tissues. We have analyzed the phenotypes of adipose-resident T cell subsets in DsbA-L^{CD4-KO} mice and control littermates fed with HFD for 5 and 12 weeks or exposed to cold. The frequencies and total numbers of CD4⁺IFN- γ ⁺ Th1 cells and CD8⁺IFN- γ ⁺T cells were both decreased in the BAT of DsbA-L^{CD4-KO} mice compared to control littermates when fed a HFD for 5 weeks (Fig.5b, c) or 12 weeks (Supplementary Fig. 5c, d). The frequencies of Treg cells in the BAT were significantly increased in DsbA-L^{CD4-KO} mice fed a HFD for 5 weeks (Fig. 5e), but were only moderately increased in the mice fed with HFD for 12 weeks (Supplementary Fig. 5f).

In response to cold stimulation, only the frequencies of IFN- γ ⁺Th1 cells were modestly decreased in the BAT of DsbA-L^{CD4-KO} mice compared to control littermates (Supplementary Fig. 6a-g). We also found that cold exposure had a much smaller effect on the frequencies of IFN- γ ⁺Th1 cells in the BAT compared to HFD feeding (Fig.5b and Supplementary Fig. 5c). However, the frequencies of Th2 subset, which was not significantly affected in the BAT by cold or HFD feeding, showed a slightly increasing trend in eWAT of DsbA-L^{CD4-KO} mice compared to floxed control mice (Fig.5d, Supplementary Fig. 5e, 6c).

Taken together, these findings suggest that DsbA-L deficiency affects the subsets of adipose-resident T cells differently, which decreased the abundance of IFN- γ -producing Th1 and CD8⁺ T cells and increased frequencies of Th2 and Treg cells in adipose tissues. These findings also suggest that DsbA-L deficiency-induced reprogramming of T cells contributes to the altered metabolic phenotypes in the DsbA-L^{CD4-KO} mice after HFD feeding. We have discussed these results in the revised manuscript (Page 10, Lines 265-270).

2. Figure 1: Does not fit on one page. For all Mitotracker/mitSOX experiments: since no individual points are plotted, N numbers have to be provided in the figure legend. Please revise. E-F) Example of the gating strategy for quantification of % mitoSOX positive cells?

Response:

We thank the reviewer for the constructive suggestion and have made changes to the figures in the revised manuscript. Data are re-arranged and are shown in column scatter diagrams. N numbers have been

provided in the figure legend. All data are now displayed on one page. Gating strategies for the quantification of mitoSOX-positive cells and MMP-low cells were shown in the revised manuscript (Fig.1c).

3. Figure 2: B) How many T cells were used for Seahorse analyses? Not stated in methods section and also not in the literature cited.

Response:

We used approximately 5×10^5 T cells per well for seahorse analyses. This information is now added in the methods section of the revised manuscript (Page 18, Lines 516-517).

4. Figure 3: B) KO mice present with reduced BW C) total fat mass is reduced in KO mice.  E) tissue weight of BAT, iWAT, pWAT reduced – also when the authors normalize it to the reduced BW? H⁺I) GTT – Please include the AUC for GTT and ITT.

Response:

We thank the reviewer for the constructive suggestions. In the revised manuscript, we have re-arranged the data by normalizing tissue weight to total body weight. Results showed that the decrease in tissue weight mainly occurred in BAT, iWAT, perirenal fat (prWAT), and liver. We have included the updated data in the revised manuscript (Fig. 3e). As suggested by the reviewer, AUC data for GTT and ITT were also included in the revised manuscript (Figures 3i, k).

5. Figure 4 – does not fit on one page. A) Quality of western blot. Beta-actin as reference protein suggests that different amounts of protein were loaded on the gel. Quantification of this blot is questionable. Nevertheless, it seems that KO mice have higher UCP1 content in BAT. D) Oxygen consumption – which group is shown? HFD or ND group? G) which tissue is shown? I) Quality of Western blot should be optimized.

Response:

- We have re-organized the data so that they fit on one page.
- To improve the quality of the data, we re-prepared samples and repeated the western blot experiments so that the beta-actin loading was more even (Fig. 4a). Through quantification of this new western blot data by Image J software, we further confirmed that UCP1 expression is increased in BAT of the HFD-fed DsbA-L^{CD4-KO} mice compared to control mice (Fig. 4a).
- (D) The HFD group of oxygen consumption is shown. We have added this information in the figure of the revised manuscript (Fig. 4d).
- (G) The oxygen consumption of BAT is shown. We have added this information in the revised manuscript. We have also added the iWAT oxygen consumption data in the revised figure (Fig. 4g).
- We have repeated the western blot experiment to improve the quality of WB shown in figure 4i of the previously submitted manuscript (Fig. 4h in the revised manuscript). We found that UCP1 expression is increased in BAT of DsbA-L^{CD4-KO} mice compared to control mice fed with HFD for 5 weeks (Fig. 4h), which is consistent with our previous finding.

6. Figure 5 – does not fit on one page.—see also related comments in Figure S5.

IFN γ levels are significantly reduced only in BAT in KO animals (B), but there are less IFN γ ⁺ CD4⁺ and also CD8⁺ T cells observed in iWAT (D-G). Here, not only the frequency but also total numbers of IFN γ ⁺ T cells are reduced in iWAT, without affecting total IFN γ protein in tissue lysates (B). Please comment.

The authors should also include eWAT in their analysis in B, since eWAT was also analyzed in D-G with significant differences and is a relevant adipose tissue depot affecting systemic metabolism.

Please stick to nomenclature for gene expression. IL-2  Il2.

Y axis – pg/ml; is it rather pg/g adipose tissue? Did the authors determine the amount of adipose tissue analyzed here volumetrically?

Response:

- We have re-arranged data so that the figure fits on one page.
- We repeated the FACS experiments by using freshly prepared samples from HFD-fed DsbA-L^{CD4-KO} mice and control littermates fed a HFD for 5 weeks. Dead cells and doublets were removed by dead-cell dye staining and FSC-A/FSC-H gating, respectively. We found that while the percentages of IFN- γ ⁺ Th1 cells in total CD4⁺ T cells were decreased, the number of these cells in iWAT was not significantly affected (Fig. 5b). It is known that iWAT contains other IFN- γ producing cells such as NK cells and ILC1 cells in addition to T cells, which are increased significantly under HFD-feeding conditions^{3,4}. The presence of these cells may mask the DsbA-L deficiency-induced alteration of IFN- γ expression in T cells in iWAT.
- As suggested by the reviewer, we have added the eWAT IFN- γ data in the revised manuscript (Fig. 5g). No significant difference in eWAT IFN- γ levels was detected between DsbA-L^{CD4-KO} mice and control mice fed with HFD for 5 weeks (Fig. 5g).
- We have checked throughout the manuscript and adopted the right nomenclature for gene expression.
- We performed the experiments according to the procedure as described previously⁵. In brief, adipose tissue (0.05g) was homogenized in 200 ul cold HBSS supplemented with proteinase inhibitors. After centrifugation at 400 g for 15 minutes at 4°C, IFN- γ levels in the supernatant of the tissue homogenates were measured by ELISA. As suggested by the reviewer, we have now presented the data in the form of IFN- γ content (ng) per gram fat tissue. We have re-organized the results in the revised manuscript (Fig.5g) and added information on the experimental procedure in the methods section of our revised manuscript (Page 16, Lines 461-465).

7. Figure 6. Y axis is now pg/ml/g fat tissue – see comment for Fig 4B. What is correct?

Is the quantification of the WB correct? Focusing on UCP1 expression in KO cells treated with IFN γ : first replicate shows nearly no UCP1 protein but strong beta-actin signal. Normalizing that, this leads to an UCP1/beta actin ration <1. In comparison to the second replicate, the related summary graph needs clarification with respect to the size of the error bar and the absolute value.

Response:

- We have changed the label of Y-axis to IFN- γ (ng) per gram fat tissue in the revised manuscript (Fig. 6a).
- The reviewer's comment on the qualification of the WB is well taken. We previously quantified the WB results from samples collected from 4 mice/group (on two gels), which showed a large sample variation. To improve the quality of the WB results, we repeated the WB experiments using samples freshly isolated from DsbA-L^{CD4-KO} mice and control mice treated with or without IFN- γ . We quantified the WB results by image J Software which showed higher UCP-1 expression in BAT of the DsbA-L^{CD4-KO} mice compared to control littermates and the increased UCP-1 expression could be suppressed by IFN- γ treatment (Fig. 7f).

Supplemental Figures

Figure S1

The authors should include the analysis of Treg frequencies to test whether thymic Treg development is impaired. The critical role of Tregs in maintaining tissue homeostasis and the response to cold was given in the introduction.

A) Quality of WB: Why is there no difference in DsbA-L protein expression in CD3⁺ T cells from floxed vs KO? It should look the same as for CD4⁺ and CD8⁺ T cells. Moreover, it should be visible in spleen (or at

least a trend) since the majority of cells in the spleen are CD4⁺ and CD8⁺ T cells. Please comment. (starting text line 130)

B) Please adhere to the nomenclature for mRNAs

Response:

- We thank the reviewer for the constructive suggestions. As suggested by the reviewer, we analyzed *Treg* frequencies by detecting the CD4⁺CD25⁺Foxp3⁺ T cells in the thymus. We found that there were no significant changes in thymic *Treg* development and have included the results in the revised manuscript (Supplementary Fig. 1f).
- To improve the quality of WB, we repeated the experiments and found that DsbA-L expression was ablated in CD3⁺T cells from DsbA-L^{CD4-KO} mice (Supplementary Fig. 1b). However, we only found a slight decrease in DsbA-L expression in the spleen of DsbA-L^{CD4-KO} mice compared to the control mice (Supplementary Fig. 1b). This slight reduction of DsbA-L expression in splenocytes was consistent with the fact that there are only about 21%-35% of the splenocytes are T cells. (<https://www.bio-rad-antibodies.com/flow-cytometry-cell-frequency.html>).
- We have made changes in the nomenclature of the mRNAs.

Figure S5

The authors should include an analysis of – at least – Foxp3⁺ Treg frequencies here (and earlier) since Tregs are known to be critically involved in regulating thermogenesis, limiting inflammation, maintaining tissue homeostasis. The fact that CD4⁺ frequencies are not altered between KO and floxed animals in the presence of altered IFN γ levels just indicates that there is likely a shift between T cell subpopulations going on. This altered immune cell composition significantly contributes to the overall metabolic effects observed and therefore, should be analyzed carefully.

Response:

We thank the reviewer for the constructive and insightful suggestions. Based on the suggestion of the reviewer, we have analyzed Foxp3⁺ *Treg* frequencies by FACS analyses in mice fed a HFD for 5 weeks and 12 weeks, respectively. We found that *Treg* cell frequencies were increased in the BAT of KO mice compared to control mice (Fig. 5e and Supplementary Fig. 5f), while Th2 cell frequencies were not obviously affected (Fig. 5d and Supplementary Fig. 5e). Since IFN- γ can suppress *Treg* differentiation^{6,7,8}, these findings are consistent with our hypothesis that the increased *Treg* frequencies may be due to decreased IFN- γ production.

Further comments:

Additionally to the WB in Figure S1, to ensure that the DsbA-L Knockout is restricted to the T cell lineage, the authors should provide a gel showing that no ectopic recombination has occurred in e.g. adipose tissues, the liver and muscles. This can be done easily by standard PCR, detecting and amplifying the recombined allele. Just to make sure that the observed effects are T cell-mediated and not due to Cre leakiness affecting major metabolic tissues.

Response: We thank the reviewer for the constructive suggestion. We have examined loxP site recombination mediated by Cre in different tissues and cell types according to the procedure as described in our previous study⁹. We detected the ~ 850bp PCR product (Supplementary Fig. 1a), an indicator of successful depletion of the DsbA-L gene in floxed DsbA-L alleles, only in CD3⁺, CD4⁺, and CD8⁺ T cells isolated from DsbA-L^{CD4-KO} mice (Supplementary Fig.1a). This result confirms that the observed effects were due to DsbA-L deletion in T cells but not in other cells caused by Cre leakage.

line 160 – The DsbA-L CD4 KO had no effect on eWAT volume or adipocyte size, but presented with reduced inflammatory gene expression (Fig 3D-G and Fig S2E). In Figure S2E there is a significant reduction in Cd3e mRNA levels in eWAT, indicating that T cell levels might be affected in the tissue upon T cell-specific DsbA-L KO. This raises the question, which T cell population is affected and whether

regulatory T cell frequencies are reduced. Detailed analyses should be included.

Response: We thank the reviewer for raising this point. Our results showed that although adipocyte size and tissue weight of eWAT were not affected, the expression of inflammatory genes was decreased to some extent (Supplementary Fig. 3a). We have examined the phenotypes of T cells in eWAT of DsbA-L^{CD4-KO} mice and their control littermates fed a HFD for 12 weeks. Flow cytometry analyses of T cell subsets (Fig. 5a) revealed that DsbA-L deficiency in T cells had little effect on T cell populations in the eWAT, as demonstrated by similar frequencies and total numbers of CD4⁺ and CD8⁺ T cells between the DsbA-L^{CD4-KO} mice and their control littermates (Supplementary Fig. 5a, b). However, the frequencies and total numbers of IFN- γ -producing Th1 cells showed a decreasing trend in the eWAT of DsbA-L^{CD4-KO} mice compared to control mice (Supplementary Fig. 5c, d). The DsbA-L^{CD4-KO} mice also showed an increasing trend in the frequencies of Treg cells, but not Th2 cells, in the eWAT compared with control littermates (Supplementary Fig. 5e, f). These results suggest that DsbA-L deficiency in T cells leads to a decrease of pro-inflammatory Th1 cells and an increase of anti-inflammatory Treg cells in the eWAT after fed a HFD for 12 weeks. Detailed analyses are included in the revised manuscript (Page 8, Lines 223 to Page 9, Lines 236).

After hypercaloric challenge (12 weeks), DsbA-L^{CD4-KO} mice present with an increase in thermogenesis in BAT. Whole body oxygen consumption and energy expenditure are increased in KO mice. Mitochondrial content and genes related to FAO are increased in BAT of KO mice. How does the long-term hypercaloric challenge affect adipose-resident T cells in KO vs floxed mice? The authors should include detailed analyses here, since the observed effects on whole body metabolism are huge keeping in mind that the authors are analyzing a T cell specific KO and therefore, the effects observed should be a result of the altered T cell population/frequency/functionality upon DsbA-L KO. The afterwards performed experiments with 5 wk HFD and IFN γ injection are not sufficient in order to reveal the underlying mechanism by which a T cell specific DsbA-L KO affects systemic metabolism in hypercaloric conditions.

Response: We greatly appreciate the valuable comments of the reviewer. To address the reviewer's question on how does the long-term hypercaloric challenge affects adipose-resident T cells in KO vs floxed mice, we examined adipose-resident T cell performances after HFD feeding for 12 weeks. Gating strategies were shown in the revised manuscript (Fig. 5a). The frequencies and total numbers of T cell subsets are shown in the revised manuscript (Supplementary Fig. 5a-f). We found that the frequencies and total numbers of CD4⁺ and CD8⁺ T cell populations were similar in adipose tissues of the HFD-fed DsbA-L^{CD4-KO} mice and control littermates (Supplementary Fig. 5a, b), but the frequencies and total numbers of IFN- γ -producing Th1 cells and IFN- γ -producing CD8⁺ T cells were significantly decreased in the BAT, and to a lesser extent in iWAT and eWAT, of DsbA-L^{CD4-KO} mice compared to control mice (Supplementary Fig. 5c, d), suggesting that DsbA-L deficiency in T cells decreased IFN- γ production after long-term hypercaloric challenge. On the other hand, the frequencies of Treg cells in the BAT were increased in DsbA-L^{CD4-KO} mice while that of Th2 cells was not affected (Supplementary Fig. 5e, f). These results indicate that DsbA-L deficiency promoted T cell reprogramming, which decreased the abundance of IFN- γ -producing T cells and increased Treg frequencies. The DsbA-L deficiency-induced T cell reprogramming may contribute to the changed metabolic phenotypes in the DsbA-L^{CD4-KO} mice. Detailed analyses are included in the revised manuscript (Page 8, Lines 223 to Page 9, Lines 236).

We agree with the reviewer that IFN- γ injection is not perfectly sufficient to reveal the underlying mechanism by which a T cell-specific DsbA-L knockout affects systemic metabolism in hypercaloric conditions. However, due to technical constraints, we were unable to upregulate IFN- γ production only in T cells of the DsbA-L^{CD4-KO} mice. It is well known that Th1 cells and CD8⁺ T cells are the main contributors of IFN- γ production in vivo. Therefore, we injected recombinant IFN- γ into mice trying to determine the in vivo role of T cell-produced IFN- γ in whole-body energy expenditure. We found that treating mice with recombinant IFN- γ led to decreased OCR and UCP1 expression in BAT of DsbA-L^{CD4-KO} mice (Fig.7e, f), suggesting that IFN- γ negatively regulates BAT thermogenic function in vivo. We have included these discussions in the revised manuscript (Page 12, Lines 323-327).

“Data not shown” – e.g. ECAR of T cells (line 134); no difference in BW, fat mass, food intake or physical activity upon IFN γ injections (line 315); line 324; line 395 no difference in IFN γ expression in BAT T cells upon cold exposure.

Response: We have included these data in the revised manuscript: The ECAR of CD4⁺ T cells and CD8⁺ T cells was not affected by DsbA-L deficiency (Supplementary Fig. 1d); IFN- γ injections did not affect body weights, fat mass, food intake, and physical activity of wild-type mice (Supplementary Fig. 8a-c); and No difference in IFN- γ expression in BAT of DsbA-L^{CD4-KO} mice and control littermates after cold stimulation (Supplementary Fig. 6h).

Line 424 – please define the exact composition of the T cell activation medium and especially define the submaximal doses of PMA and ionomycin. Was the media supplemented with pyruvate, NEA, etc.?

Response: The T cell activation medium is the 1640 medium containing 10% FBS, L-Glutamine (1mM), β -Mercaptoethanol (50 μ M), and a submaximal dose of PMA (5 ng/ml) and ionomycin (75 ng/ml). We did not add extra pyruvate or NEA to the medium. We have added this information in the Methods section (Page 16, Lines 455-457).

The authors should provide a full gating strategy in the supplement, as they did it in Fig. 4H. Otherwise, it is difficult to evaluate flow cytometric analyses.

Did the authors use a dead cell stain? This is especially important when analyzing immune cells isolated from adipose tissues – since the isolation protocol itself, as well as the lipid rich environment during isolation easily kills immune cells.

Response: As suggested by the reviewer, we have added gating strategies of mitoSOX-positive and MMP-low cells in Fig. 1c, fat-resident T cell subsets in Fig.5a, fat-resident $\gamma\delta$ T cells in Supplementary Fig.5g, and fat-resident M2 and eosinophils in Supplementary Fig. 5i in the updated manuscript. We have repeated the FACS-related experiments and excluded dead cells through dead cell dye (FVS-520 or Zombie blue). Results showed that after excluding dead cells, HFD promoted MMP^{low} and mitoSOX-positive T cells in the BAT but not iWAT. DsbA-L deficiency in T cells decreased the frequencies of IFN- γ -producing T cells while promoting Treg accumulation in the BAT after HFD feeding for 5 and 12 weeks, while the percentages of fat-resident $\gamma\delta$ T cells, M2, and eosinophils were not affected. These results are consistent with our previous findings.

As depicted in Fig. 5H, it seems that no doublet exclusion (and no live/dead cell discrimination, as mentioned above) was performed, which is critical to eliminate readout errors.

Response: The reviewer’s comments are well-taken. We have re-conducted the FACS experiments and excluded dead cells through dead cell dye. Also, we have performed a doublet discrimination experiment using a pulse geometry gate (FSC-H x FSC-A). We have added gating strategies in Fig. 1c, Fig. 5a, Supplementary Fig. 5g, and Supplementary Fig. 5i in the updated manuscript. Results showed that after excluding dead cells and doublets, HFD promoted MMP^{low} and mitoSOX-positive T cells in the BAT but not iWAT. DsbA-L deficiency in T cells decreased the frequencies of IFN- γ -producing T cells while promoting Treg accumulation in the BAT after HFD feeding for 5 and 12 weeks, while the percentages of fat-resident $\gamma\delta$ T cells, M2, and eosinophils were not affected. These results are consistent with our previous conclusions.

Related to the methods section and specifically the isolation of T cells – CD3⁺ and CD8⁺ T cells were isolated using PE-labeled antibodies and anti-PE beads, which is fine. Then the isolated population contains both, CD4⁺ and CD8⁺ T cells. While CD4⁺ T cells were isolated separately, but not first, as it seems. For which experiments have these enriched cells been used?

Response: We isolated CD4⁺, CD3⁺, and CD8⁺ T cells from spleen separately. We used anti-CD4 magnetic

beads to isolate CD4⁺ T cells. For CD3⁺ and CD8⁺ T cells, we first used PE-labeled antibodies to label CD3⁺ or CD8⁺ cells from spleen, respectively, and then applied to anti-PE magnetic beads to isolate them. These CD3⁺, CD4⁺, and CD8⁺ T cells were used in the T cell experiments in Fig. 2 and Supplementary Fig. 1. The information is included in the methods section of the revised manuscript (Page 19, Lines 560 to Page 20, Lines 567).

Line 547 – which post-hoc test was used to adjust for multiple comparison? Please clarify.

Response: Multiple comparisons between the groups were performed using Tukey’s test. We have included this information in the methods section.

The authors should carefully revise their figure legends with regard to

1. Include N numbers for all bar graphs

2. Check for spelling mistakes (some are listed below) and please, adhere to nomenclature

Text related

Line 53 – relied (rely)

Abbreviation Treg not introduced

Line 102 – Is T cell mitochondrial mass really a biomarker for T cell biogenesis? Consider revising

Line 106 – m ice  mice

Line 107 – to a lesser extent

Line 145 – (

Line 364 - ... by which DsbA-L deficiency increases Treg expression remains unknown.

Figure legends - °C missing (e.g. figure 4); several times gourp  group

Response: We thank the reviewer for these constructive suggestions. We have added scatter plots to our bar-graphs. N numbers were added when necessary. We have corrected typos and grammar errors.

- We have changed “relied” into “rely”, “m ice” into “mice”, “less extent” into “lesser extent”, “gourp” into “group” in the revised manuscript.
- We have introduced the full name of Treg as regulatory T cells in the revised manuscript (Page 3, Line 55).
- We have rephrased the expression as “T cell mitochondrial mass, a biomarker for T cell mitochondrial function” in the revised manuscript (Page 4, Lines 86-87).
- We have changed the word “expression” into “expansion” in the revised manuscript in the revised manuscript (Page 14, Line 381).
- We have added “°C” after the temperatures in the revised manuscript.

Reviewer #2:

Fig1C-D. What shown in Fig 1C-D seems to be the opposite of what is described in the text.

It seems that the HFD increases MMP in CD4⁺ and CD8⁺ T cells in BAT (Fig 1C) and decreases MMP in CD4⁺ and CD8⁺ T cells in iWAT (Fig 1D).

Response: Fig.1C showed increased (gated) MMP^{low} cells in the BAT, which indicates decreased mitochondrial membrane potential (MMP) and impaired mitochondrial function. We found that HFD decreased MMP in CD4⁺ and CD8⁺ T cells in the BAT (Fig. 1d) and increases MMP in CD4⁺ and CD8⁺ T cells in the iWAT (Fig. 1e), suggesting that overnutrition has a distinct effect on T cell mitochondrial function in mouse BAT and iWAT. Given that the total numbers and diversities of immune cells are much less in BAT than that in iWAT¹⁰, a possible explanation for the above results is that T cells and other immune cells may be more sensitive to HFD-induced stress in BAT than in iWAT, leading to different MMP levels in BAT and iWAT. We have discussed this possibility, which may be addressed in future studies, in our revised manuscript (Page 5, Lines 101-103).

To specifically delete DsbA-L in T cells, the authors use a CD4-Cre mouse. If the Cre mouse is CD4⁺

specific, how it is possible that there is deletion also in CD8⁺ and CD3⁺ cells? Can the authors comment on that?

Response: We appreciate the comments of the reviewer. CD4⁺ and CD8⁺ T cell precursors go through a CD4⁺CD8⁺ double-positive stage during the development in the thymus, then they make a lineage decision to become CD4⁺ or CD8⁺ T cells¹¹. During this double-positive stage, CD4-mediated Cre will be expressed, which will cause DsbA-L deletion in this CD4⁺CD8⁺ double-positive T cell precursors and subsequent CD4⁺ T cells and CD8⁺ T cells. CD3⁺ T cells are mainly composed of CD4⁺ T cells and CD8⁺ T cells, so the DsbA-L deletion can also be detected in the isolated CD3⁺ T cells. This phenomenon was frequently reported in previous models using CD4 cre¹². We have added the information on the knockout strategy in the Method section of the revised manuscript (Page 16, Lines 443-446)

Fig2B. DsbA-L depletion in T cells leads to a decrease in basal and maximal respiration in CD4⁺ cells and only a decrease in the max respiration in CD8⁺. Do the authors have an explanation for this difference? What happened to CD3⁺ cells?

Response:

- Basal respiration is comprised of three parts: ATP-Linked OXPHOS, proton leak, and non-mitochondrial respiration. We did not observe any significant changes in proton leak either between DsbA-L-deficient CD4⁺ T cells and wild-type CD4⁺ T cells or between DsbA-L-deficient CD8⁺ T cells and wild-type CD8⁺ T cells (Fig.2b). It is known that activation of both CD8⁺ and CD4⁺ T cells rapidly switches the metabolic programs from OXPHOS toward aerobic glycolysis^{13, 14}. CD8⁺T cells may rely more on aerobic glycolysis but less on OXPHOS for energy supply compared to CD4⁺ T cells. Consistently, glucose has been shown to be the most critical source of energy for CD8⁺ effector T cells to execute critical immune functions such as anti-viral and anti-tumor immune response¹⁵. Therefore, when DsbA-L-deficient CD4⁺ T cells showed an obvious reduction in basal OCR compared to wild-type CD4⁺ T cells, DsbA-L-deficient CD8⁺ T cells displayed a less obvious decrease in the basal OCR compared to wild-type CD8⁺ T cells, suggesting that OXPHOS in CD8⁺ T cells was less affected by DsbA-L deficiency. However, we could not exclude the possibility that DsbA-L deficiency in T cells may decrease non-mitochondrial respirations in CD4⁺ T cells but not in CD8⁺ T cells. Future studies are needed to test the hypothesis. Nevertheless, the findings that DsbA-L deficiency led to a decreasing trend in basal OCR and a significant decrease in maximal respiration in CD8⁺ T cells still support the conclusion that DsbA-L is essential for optimal mitochondrial respiration of T cells.
- As to CD3⁺ T cells, we found that DsbA-L deficiency in CD3⁺ T cells impaired both basal and maximal respiration (Supplementary Fig. 1c). This finding suggests that DsbA-L is essential for optimal CD3⁺ T cell mitochondrial function.

FigS2E. The authors suggest that the DsbA-L T cells specific ko mouse present reduced inflammation in eWAT vs controls. In this panel, only one inflammatory gene is shown to be reduced, the authors should provide more evidence of a reduced inflammation.

Response: We thank the reviewer for this constructive suggestion. Based on the suggestion, we have added more gene expression profiles to further show the inflammation state of eWAT in DsbA-L deficient mice and control mice. Our results showed that inflammatory genes such as *Ifng*, *Il17*, and *Cd11c* were decreased to some extent in eWAT of DsbA-L^{CD4-KO} mice compared with control littermates (Supplementary Fig. 3a), further demonstrating that inflammation is reduced in eWAT of the HFD-fed DsbA-L deficient mice compared to wild-type control mice.

Fig3H-I. The authors should normalise the GTT data to the basal and provide the AUC. Moreover, what are the values of fasting glucose and insulin of the ko mice vs control?

Response: We thank the reviewer for this constructive suggestion. The required AUC figures for GTT and

ITT were now included in the revised manuscript (Fig. 3i, k). AUC data of GTT showed better glucose tolerance in KO mice, AUC data of ITT showed better insulin sensitivity in KO mice, which is consistent with the finding from our previous study. After HFD feeding for 12 weeks, DsbA-L^{CD4-KO} mice showed decreased fasting glucose levels without affecting fasting insulin levels. Data are included in the revised manuscript (Fig. 3h, j).

Fig4D-F. The authors should normalize the indirect calorimetry data by the lean mass of the mice or using an ancova test.

Response: We thank the reviewer for this constructive suggestion. For data processing, we used CalR (an abbreviated form of calor, the Latin word for heat), a web-based analysis tool to normalize indirect calorimetry data¹⁶. ANCOVA analyses of average VO₂ (ml/hr) with statistically controlling for the effects of covariate lean mass were included in the revised manuscript (Supplementary Fig. 3i).

Fig4G-H. Is the basal OCR of iWAT increased too? The authors should provide this data. Moreover, how is the expression of cpt1 and ppara in the iWAT?

Response:

- The basal OCR of iWAT is increased in HFD-fed DsbA-L^{CD4-KO} mice compared with control littermates (Fig. 4f). This result is consistent with an improved mitochondrial function in the iWAT of DsbA-L^{CD4-KO} mice as indicated by enhanced expression of genes related with mitochondrial function, such as *Atp5a1*, *Acox1* (Fig. 4g), and *Tbx1* (Fig. 4i).
- There is an increasing trend in the expression of *Cpt1* and *Ppara* genes in iWAT of DsbA-L^{CD4-KO} mice compared to control littermates, but it did not reach statistical significance (Fig. 4g).

Fig4J. CD137 cannot really be considered anymore as a beige marker (see paper "CD137 negatively affects "browning" of white adipose tissue during cold exposure." Srivastava RK, Moliner A, Lee ES, Nickles E, Sim E, Liu C, Schwarz H, Ibáñez CF. J Biol Chem. 2020 Feb 14;295(7):2034-2042. doi: 10.1074/jbc.AC119.011795.).

Response: We thank the reviewer for informing us of this new finding and have removed the data regarding *Cd137* in our revised manuscript (Fig. 4i).

Fig4K-M. The authors should normalize the indirect calorimetry data by the lean mass of the mice or using an ancova test. Same for Fig7B and C.

It would be interesting if the authors could show also that the BAT of the ko mice vs controls show increased max respiration capacity after norepinephrine treatment at thermoneutrality.

Response:

- We thank the reviewer for this constructive suggestion. Based on the suggestion of the reviewer, we normalized our data in Fig. 4K-M and calculated the statistical significance using ANCOVA analyses of average VO₂ (ml/hr) by lean mass. The results are added in the revised manuscript (Supplementary Fig. 3j).
- For Fig.7B-7C, ANCOVA analyses of average VO₂ (ml/hr) by lean mass were included in the revised manuscript (Fig. 7d).
- We thank the reviewer for the suggestion on the norepinephrine experiment. However, based on the finding that cold exposure had no significant effect on IFN- γ protein and mRNA levels in the BAT between DsbA-L^{CD4-KO} mice and control littermates (Supplementary Fig. 4a-c), it is most likely that norepinephrine treatment at thermoneutrality has no significant effect on maximal respiration capacity between DsbA-L^{CD4-KO} mice and control littermates. We are unable to directly test this possibility due to the close of the Wuhan University Metabolic Lab due to the COVID-19 epidemic.

Fig.S4. Why the DbsA-L ko mice have increased DIT and not CIT? what is the explanation for that? Can the authors comment on it?

Response: It is known that lean mice have very low levels of IFN- γ in adipose tissue^{17, 18, 19} and cold exposure could further inhibit IFN- γ mRNA expression in the BAT (Supplementary Fig. 6h). These results are consistent with the finding that cold exposure suppresses immune responses in adipose tissues^{20, 21, 22}. IFN- γ gene expression in the BAT of the DsbA-L^{CD4-KO} mice did not differ between room temperature and cold exposure conditions (Supplementary Fig. 6h), probably because IFN- γ expression was already reduced to its basal level in the BAT. Another possible explanation may be that the promoting effect of cold exposure on BAT thermogenesis is much more dominant than HFD feeding, so it might cover up the difference in CIT between DsbA-L^{CD4-KO} mice and floxed control mice. We have discussed these possibilities in the revised manuscript (Page 15, Lines 406-418).

Fig5. The authors should provide the same data shown in this figure for BAT, iWAT, eWAT of mice exposed to cold, to show that the effect is specific for DIT.

Response: We thank the reviewer for this constructive suggestion. Based on the suggestion, we have provided new data on BAT, iWAT, eWAT of mice exposed to cold. We found that only the frequencies of IFN- γ ⁺Th1 cells decreased modestly in the BAT of DsbA-L^{CD4-KO} mice compared to control littermates (Supplementary Fig. 6a-g). Results also showed that the frequencies of IFN- γ ⁺Th1 cells in the BAT of cold-stimulated mice were much lower than that of HFD-fed mice (Fig.5b and Supplementary Fig. 5c). These results provide an explanation as to why cold exposure-induced thermogenesis was comparable between DsbA-L^{CD4-KO} mice and control littermates.

If the total amount of serum IFN γ is unchanged in the ko mice vs controls on HFD, how is possible the effect of increased DIT is due to a decrease in IFN γ in vivo? The authors should provide an explanation for that.

Response: We appreciate the reviewer's valuable comments. HFD feeding induces a low-grade inflammatory state at which serum IFN- γ levels vary little^{17, 18, 19}. This chronic low-grade inflammation is different from acute infectious disease-induced strong systemic inflammation that causes a significant increase of IFN- γ levels in the serum²³. In addition, BAT-resident T cells account for only a small proportion of T cells in the whole body. Other IFN- γ -producing cells, such as NK cells, will also contribute to the serum IFN- γ levels. We believe that inflammation in BAT of HFD-fed mice is of low-grade and is tissue-restricted. This low-grade local inflammatory change will affect the metabolism of BAT but does not cause strong inflammatory changes throughout the body. We have discussed this possibility in the revised manuscript (Page 15, Lines 419-429).

Fig.7 How physiological is the amount of IFN γ given to the mice? How was the amount to be injected calculated? Is that amount similar to the one measured in the serum of wt mice after HFD challenge?

Response: The amount of IFN- γ given to the mice, while is higher than the serum levels of IFN- γ in HFD-fed mice (about 25 pg/ml), is consistent with the dose used in previous studies^{6, 24}. Studies have revealed that mice treated with 2-10 μ g IFN- γ could suppress adipogenesis or adipose tissue homeostasis^{6, 24}. To avoid unwanted side effects by a higher dose of IFN- γ administration, we thus adopted the dose of 2 μ g IFN- γ per mice. Given that mice in these experiments were normally between 20 and 25g, the final dose of IFN- γ was thus ~100 μ g/kg.

Minor comments

Line 145 of the text. The authors should remove the “(“.

Response: We thank the reviewer for pointing out our negligence and have corrected the mistake in the revised manuscript.

Reviewer #3:

1. In Fig 3H and I, the GTT and ITT experiments were applied to detect glucose tolerant and insulin sensitive, whether phosphorylated protein expression (PKB and GSK3 β) can be detected to represent insulin resistance? Please refer (Diabetes 2018 04;67(4)).

Response: We thank the reviewer for this constructive suggestion. Based on the reviewer's suggestion, we performed experiments to examine insulin signaling in HFD-fed DsbA-L^{CD4-KO} mice and control littermates. In brief, mice were fasted overnight and injected with 0.75U/kg insulin intraperitoneally. Fat tissues were collected 10 minutes later. Phosphorylation of AKT (another name of PKB) on Ser308 and Ser473 and GSK3 β on Ser9 were examined by WB according to a similar procedure as described²⁵. We found that insulin sensitivity was improved in all the fat tissues of the DsbA-L^{CD4-KO} mice compared to control mice (Supplementary Fig. 3b-d).

2. In Fig 2, the author proved that DsbA-L was a critical regulator of T cell credible function, whether DsbA-L deficiency affect mitochondrial dynamics in T cell?

Response: We thank the reviewer for this question. By detecting the mitochondrial fission marker DRP1 and fusion marker OPA1 and MFN2, we found that DsbA-L deficiency increased mitochondrial fission after activation while mitochondrial fusion was not significantly affected (Fig. 2e). By staining T cells with Mito-tracker Green, we also found a slight increase of mitochondrial fragmentation in DsbA-L-deficient T cells than wild type T cells (Fig. 2f). Given that fused mitochondrial is beneficial for OXPHOS²⁶, these results further support our conclusion that DsbA-L deficiency decreased OXPHOS in T cells.

3. In the study, the author detected the function of mitochondria, such as OCR, ATP and mtDNA. What`s the potential role of calcium ions within mitochondria in these changes under the condition of DsbA-L deficiency? The authors should discuss this point.

Response: By using a specific mitochondrial calcium dye Rhod2, we examined the kinetics of calcium signal changes in different cells over time. We found that DsbA-L deficiency suppressed TCR-stimulated mitochondrial calcium levels in activated CD4⁺ T cells (Fig. 2g). Given that calcium is important for mitochondrial respiration²⁷, this result further supports a key role of DsbA-L in regulating mitochondrial function in T cells.

4. In Fig4, Which subtype of CPT1 is it? CPT1A, CPT1B or CPT1C?

Response: It is CPT1a. We have added information in the revised manuscript (Page 8, Line 197) and Fig.4g.

References:

1. Mauro C, *et al.* Obesity-Induced Metabolic Stress Leads to Biased Effector Memory CD4(+) T Cell Differentiation via PI3K p110delta-Akt-Mediated Signals. *Cell Metab* **25**, 593-609 (2017).
2. Han SJ, *et al.* White Adipose Tissue Is a Reservoir for Memory T Cells and Promotes Protective Memory Responses to Infection. *Immunity* **47**, 1154-1168 e1156 (2017).
3. Boulouvar S, *et al.* Adipose Type One Innate Lymphoid Cells Regulate Macrophage Homeostasis through Targeted Cytotoxicity. *Immunity* **46**, 273-286 (2017).
4. Wensveen FM, *et al.* NK cells link obesity-induced adipose stress to inflammation and insulin resistance. *Nat Immunol* **16**, 376-385 (2015).

5. Moysidou M, *et al.* CD8+ T cells in beige adipogenesis and energy homeostasis. *JCI Insight* **3**, (2018).
6. Molofsky AB, *et al.* Interleukin-33 and Interferon-gamma Counter-Regulate Group 2 Innate Lymphoid Cell Activation during Immune Perturbation. *Immunity* **43**, 161-174 (2015).
7. Deng T, *et al.* Adipocyte adaptive immunity mediates diet-induced adipose inflammation and insulin resistance by decreasing adipose Treg cells. *Nature Communications* **8**, 15725 (2017).
8. Zhu J, Paul WE. CD4 T cells: fates, functions, and faults. *Blood* **112**, 1557-1569 (2008).
9. Chen H, *et al.* Hepatic DsbA-L protects mice from diet-induced hepatosteatosis and insulin resistance. *FASEB J* **31**, 2314-2326 (2017).
10. Prunet-Marcassus B, Cousin B, Caton D, Andre M, Penicaud L, Casteilla L. From heterogeneity to plasticity in adipose tissues: site-specific differences. *Exp Cell Res* **312**, 727-736 (2006).
11. Germain RN. T-cell development and the CD4-CD8 lineage decision. *Nat Rev Immunol* **2**, 309-322 (2002).
12. Sharma S, Zhu J. Immunologic applications of conditional gene modification technology in the mouse. *Curr Protoc Immunol* **105**, 10.34.11-10.34.13 (2014).
13. Chang CH, *et al.* Posttranscriptional control of T cell effector function by aerobic glycolysis. *Cell* **153**, 1239-1251 (2013).
14. Konjar S, Veldhoen M. Dynamic Metabolic State of Tissue Resident CD8 T Cells. *Front Immunol* **10**, 1683 (2019).
15. Cham CM, Driessens G, O'Keefe JP, Gajewski TF. Glucose deprivation inhibits multiple key gene expression events and effector functions in CD8+ T cells. *Eur J Immunol* **38**, 2438-2450 (2008).
16. Mina AI, LeClair RA, LeClair KB, Cohen DE, Lantier L, Banks AS. CalR: A Web-Based Analysis Tool for Indirect Calorimetry Experiments. *Cell Metab* **28**, 656-666 e651 (2018).
17. Schmidt FM, *et al.* Inflammatory cytokines in general and central obesity and modulating effects of physical activity. *PLoS One* **10**, e0121971 (2015).
18. Nehete P, Magden ER, Nehete B, Hanley PW, Abee CR. Obesity related alterations in plasma cytokines and metabolic hormones in chimpanzees. *Int J Inflam* **2014**, 856749 (2014).
19. Azizian M, *et al.* Cytokine profiles in overweight and obese subjects and normal weight individuals matched for age and gender. *Ann Clin Biochem* **53**, 663-668 (2016).
20. Vargovic P, Manz G, Kvetnansky R. Continuous cold exposure induces an anti-inflammatory response in mesenteric adipose tissue associated with catecholamine production and thermogenin expression in rats. *Endocr Regul* **50**, 137-144 (2016).
21. Reynes B, van Schothorst EM, Garcia-Ruiz E, Keijer J, Palou A, Oliver P. Cold exposure down-regulates immune response pathways in ferret aortic perivascular adipose tissue. *Thromb Haemost* **117**, 981-991 (2017).

22. Vargovic P, Laukova M, Ukropec J, Manz G, Kvetnansky R. Prior Repeated Stress Attenuates Cold-Induced Immunomodulation Associated with "Browning" in Mesenteric Fat of Rats. *Cell Mol Neurobiol* **38**, 349-361 (2018).
23. Lauw FN, *et al.* Elevated plasma concentrations of interferon (IFN)-gamma and the IFN-gamma-inducing cytokines interleukin (IL)-18, IL-12, and IL-15 in severe melioidosis. *J Infect Dis* **180**, 1878-1885 (1999).
24. Vidal C, Bermeo S, Li W, Huang D, Kremer R, Duque G. Interferon gamma inhibits adipogenesis in vitro and prevents marrow fat infiltration in oophorectomized mice. *Stem Cells* **30**, 1042-1048 (2012).
25. Tubbs E, *et al.* Disruption of Mitochondria-Associated Endoplasmic Reticulum Membrane (MAM) Integrity Contributes to Muscle Insulin Resistance in Mice and Humans. *Diabetes* **67**, 636-650 (2018).
26. Buck MD, *et al.* Mitochondrial Dynamics Controls T Cell Fate through Metabolic Programming. *Cell* **166**, 63-76 (2016).
27. Kaufmann U, Kahlfuss S, Yang J, Ivanova E, Korolov SB, Feske S. Calcium Signaling Controls Pathogenic Th17 Cell-Mediated Inflammation by Regulating Mitochondrial Function. *Cell Metab* **29**, 1104-1118 e1106 (2019).

Reviewers' Comments:

Reviewer #1:

Remarks to the Author:

In the opinion of this reviewer, the authors have carefully addressed several of the concerns raised by the reviewers. They revised the manuscript and figures and provided additional data and evaluations to support their findings.

In the view of this reviewer, there is sufficient progress of the presented findings to support publication in Nature Communications.

Reviewer #2:

Remarks to the Author:

We are satisfied of the way the authors addressed our comments. We consider that the new version is suitable for publication in Nat comm.

Reviewer #3:

Remarks to the Author:

Thanks to the authors for their perfect answers, which perfectly solved my confusion.

REVIEWERS' COMMENTS

Reviewer #1 (Remarks to the Author):

In the opinion of this reviewer, the authors have carefully addressed several of the concerns raised by the reviewers. They revised the manuscript and figures and provided additional data and evaluations to support their findings.

In the view of this reviewer, there is sufficient progress of the presented findings to support publication in Nature Communications.

Response: We thank the reviewer for the constructive comments and kind evaluation.

Reviewer #2 (Remarks to the Author):

We are satisfied of the way the authors addressed our comments. We consider that the new version is suitable for publication in Nat comm.

Response: We appreciate the kind comments of the reviewer.

Reviewer #3 (Remarks to the Author):

Thanks to the authors for their perfect answers, which perfectly solved my confusion.

Response: We appreciate the kind comments of the reviewer.